# Intergenerational effects of a casino-funded family transfer program on educational outcomes in an American Indian community

Tim A. Bruckner [1,2] ✉, Brenda Bustos[2], Kenneth A. Dodge[3],
Jennifer E. Lansford [3], Candice L. Odgers [4] & William E. Copeland [5]

Cash transfer policies have been widely discussed as mechanisms to curb intergenerational transmission of socioeconomic disadvantage. In this paper, we take advantage of a large casino-funded family transfer program introduced in a Southeastern American Indian Tribe to generate difference-in-difference estimates of the link between children's cash transfer exposure and third grade math and reading test scores of their offspring. Here we show greater math (0.25 standard deviation [SD], $p$ =.0148, 95% Confidence Interval [CI]: 0.05, 0.45) and reading (0.28 SD, $p$ = .0066, 95% CI: 0.08, 0.49) scores among American Indian students whose mother was exposed ten years longer than other American Indian students to the cash transfer during her childhood (or relative to the non-American Indian student referent group). Exploratory analyses find that a mother's decision to pursue higher education and delay fertility appears to explain some, but not all, of the relation between cash transfers and children's test scores. In this rural population, large cash transfers have the potential to reduce intergenerational cycles of poverty-related educational outcomes.

Parents' wealth plays a substantial role in their children's life chances[1,2]. In the United States, 13 million children live in families with incomes below the poverty line[3]. Extensive literature finds that these children show an increased risk of poor physical and cognitive outcomes[4-9] as well as lower socioeconomic status attainment in adulthood[10,11]. Increasing recognition of the strong intergenerational transmission of disadvantage, and the relatively high fraction of children living in poverty in the US[12], has led to a variety of interventions which aim to improve life outcomes for low-income children. Some scholars and policymakers, for instance, have proposed direct cash transfers (e.g., a child tax credit) to boost the financial resources of low-income families with children[12-14].

Accumulating evidence[15], including from the Great Smoky Mountains Study (GSMS) in rural North Carolina which began recruitment before a "natural experiment," supports causal long-term

benefits of a large family cash transfer during childhood. In the late 1990s, a Southeastern American Indian Tribe underwent a natural experiment by way of the introduction of a casino on their lands. Under the terms of an agreement with the tribe, the casino allocated a percentage of profits in acute lump sums to all enrolled members. Gaming proved profitable; since 1996, per capita payments to members have averaged approximately $5000 per year. These disbursements raised income levels of an entire community that previously exhibited a high rate of poverty. GSMS findings indicate improved educational attainment[13], health[16] and financial well-being into adulthood among American Indian participants whose families received cash transfers during their childhood[17]. Importantly, findings appear stronger with increasing duration of time that their American Indian families received the transfers while the child lived at home[17]. This result coheres with work in economics which finds that early childhood

[1]Center for Population, Inequality, and Policy, University of California, Irvine, CA 92697, USA. [2]Joe C. Wen School of Population & Public Health, University of California, Irvine, CA 92697, USA. [3]Sanford School of Public Policy, Duke University, Durham, NC 27708, USA. [4]Department of Psychological Science, University of California, Irvine, CA 92697, USA. [5]Department of Psychiatry, University of Vermont, Burlington, VT 05405, USA. ✉e-mail: tim.bruckner@uci.edu

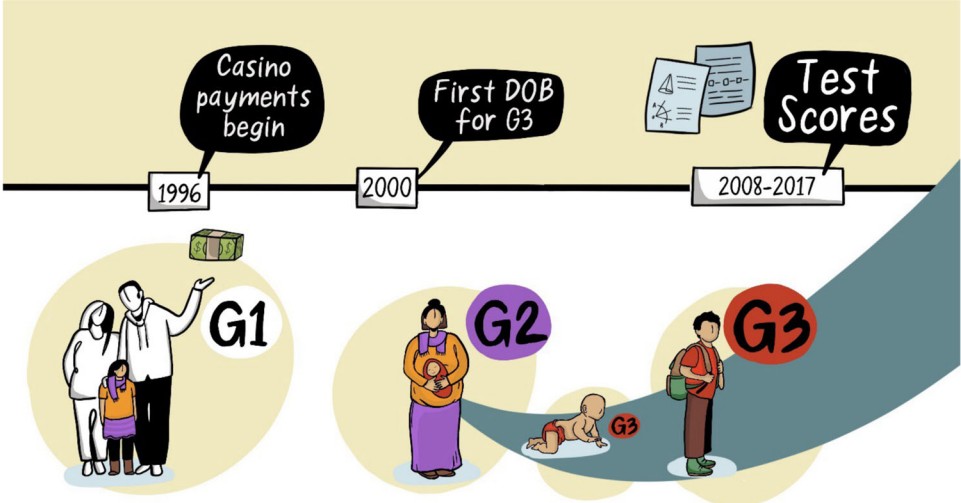

**Fig. 1 | Timeline of casino payments to American Indian families and third grade test scores of the next generation of American Indian children.** Casino payments begin in 1996 and are disbursed to adults (G1). The young children of G1 (i.e., G2) grow to childbearing age, and 2000 is the first birth year of their children (G3) for whom we retrieved third grade reading and math test scores from 2008 to 2017. G2 women who were relatively young in 1996 –when G1 received the first Casino payment– are considered more exposed to the cash transfers than are G2 women who were at or above 18 years of age in 1996.

investments offer greater long-term gains to human capital than do investments later in life[18].

Whereas many interventions aim to improve outcomes for low-income children, few examine whether their effects persist into the next generation. In this study, we exploit the quasi-random timing of the cash transfer during childhood among the tribe to test whether the next generation of children show human capital gains. We use the second generation's math and reading test score data in third grade—a reliable predictor of later-life educational attainment[19,20] and the earliest year in which standardized educational outcomes are obtained—as a gauge of intergenerational effects. In addition, unlike earlier work, we focus on the population base of American Indians that had children (rather than a selected cohort) which permits not only increased study power but also external validity to the affected region.

In this work, we use American Indian race/ethnicity as a proxy for tribal membership and find improved third grade math and reading scores among American Indian students whose mother was exposed longer to the cash transfer during her childhood. A mother's decision to pursue higher education and delay fertility explains some, but not all, of the discovered relation. In this rural population, large cash transfers have the potential to enhance human capital of the next generation.

## Results

### Exposure and sample characteristics

Consistent with prior work, we used American Indian race/ethnicity in Jackson, Swain, and Graham counties in North Carolina as a proxy for the Eastern Band of Cherokee. These residents received the large family cash transfer beginning in 1996. By contrast, non- American Indian residents in these counties received no cash transfer. Figure 1 provides a timeline of the cash transfers to American Indian families, the timing of births, and the data linkage to third grade test scores.

Using state administrative records housed at the North Carolina Education Research Data Center (NCERDC), we accessed the linked North Carolina Birth file to math ($N = 4289$) and reading test scores ($N = 4254$) for third grade public school students in the three treated counties, from 2008 to 2017. Whereas mean scores for non- American Indian children ($N = 3549$) lie slightly above the state mean, those for American Indian children ($N = 740$) fall, on average, 0.39 standard deviations (SD) below the state mean (Fig. 2).

Table 1 describes maternal and birth characteristics of the children with valid third grade test scores. American Indian mothers tend to report lower completed education, younger age at birth, and lower frequency of being married than do non- American Indian mothers. By contrast, the prevalence of preterm (<37 weeks completed gestational age at delivery) and/or low weight (<2500 g) delivery is lower among births to American Indian mothers (vs. non-American Indian mothers). These patterns appear consistent with the broader literature describing racial/ethnic differences, which indicates minimal bias in the NCERDC algorithm used to link birth records to third grade test scores.

### Regression for third grade math and reading scores

We employed a "difference-in-difference" (DiD) regression strategy to isolate potential benefits of the family cash transfer on educational outcomes of children born to American Indian mothers who were relatively young in 1996—the first year of the family cash transfer program. This approach uses two control populations (e.g., non-American Indian children as well as children born to American Indian mothers who were relatively older, around age 17 in 1996) to adjust for unmeasured confounding and other threats to validity. Our DiD specification is a time-varying treatment effect design in which duration of exposure to the cash transfer as a child serves as the "intensity" of exposure for American Indian mothers[21]. We examine the influence of the cash transfer by regressing children's test scores on time exposed to the cash transfer and American Indian status, and then testing whether the relation between test scores and time exposed to the cash transfer differs by American Indian status. Here, duration of time exposed before age 18 is a continuous variable (range: 0–15 years; see Supplementary Table 1). A person over 18 in 1996 receives a "0" duration value and we retain them in the sample. Importantly, our dataset also permits a test of the parallel trends assumption in the pre-treatment period (see Supplementary Tables 2 and 3).

Results from the DiD regressions (Model 1 column in Table 2 for Math; Model 1 column in Table 3 for Reading) show a positive relation between test scores and the interaction term of American Indian race/ethnicity and childhood remaining at the start of the family cash transfer. The positive relation reaches conventional levels of statistical detection (i.e., $p < 0.05$) for both reading ($p = 0.0014$, 95% CI: 0.013, 0.055) and math ($p = 0.0055$, 95% CI: 0.009, 0.050) scores.

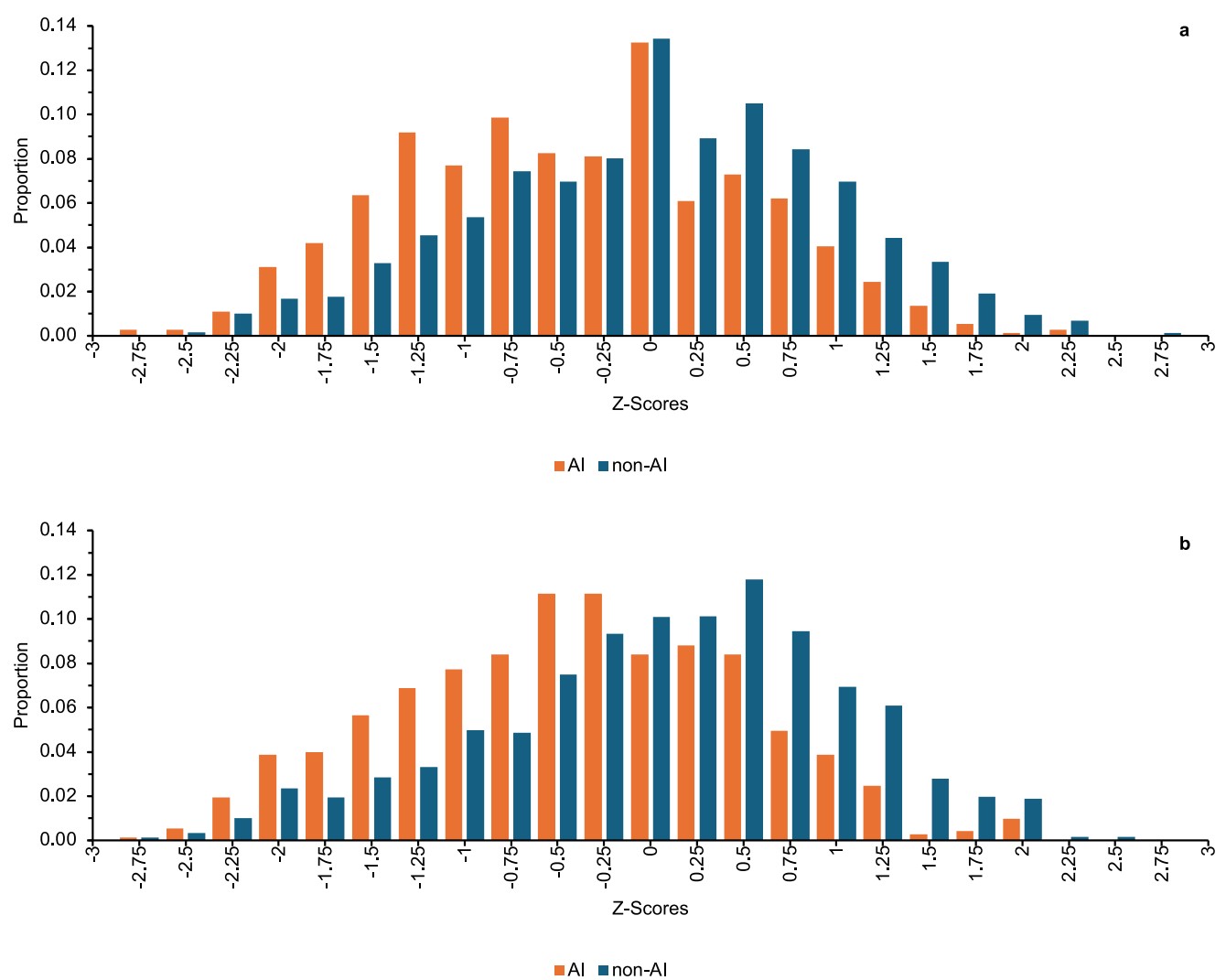

**Fig. 2 | Histograms of the Z-Scores of Third grade Tests.** Third grade math (**a**) and reading (**b**) scores among children born to American Indian (orange bar) and non-American Indian mothers (blue bar), Jackson, Swain, and Graham counties, for test years 2008–2017. The orange bars represent the proportion of a z-score to children of American Indian mothers. The blue bars represent the proportion of a z-score to children of non- American Indian mothers.

The strength of the relation is slightly larger (i.e., ~17%) for reading relative to math. Inclusion of child age at time of test (with a squared and cubed term; see Model 2 column in Tables 2 and 3) and infant sex slightly attenuates main results but does not affect statistical inference. Figure 3 (math) and Figure 4 (reading) illustrate the regression results of Model 2 in Tables 2 and 3 by showing fitted third grade test scores by American Indian status and category of duration exposure. Within the context of the declining trend in third grade test scores in this rural population (which mirrors national trends[22]), the race-based disparity in test scores narrows for American Indian children whose mothers had a relatively greater duration exposure.

**Summary of findings**

To give the reader a sense of the magnitude of the findings, a child whose American Indian mother with ten years of exposure to the family cash transfer before age 18 years scores 0.25 SD higher in math, and 0.28 SD higher in reading, relative to a child whose American Indian mother had no exposure to the family cash transfer before age 18 years (per coefficients in Model 2 column). This value, while smaller than the observed American Indian/non- American Indian gap in test scores at third grade, is greater than the average score gap between a child whose mother graduated from high school and a child whose

mother did not graduate from high school. This value is similar in magnitude to $1000 per pupil per year investments in early childhood education interventions in North Carolina[23]. When scaled to other early childhood educational interventions[23], the magnitude of the test score increases equates to an additional half school year of learning. Furthermore, these results appear consistent with a continued educational benefit, of moderate magnitude, that affects not only the generation of parents (G2; see Akee et al.[13]) but also their children.

The discovered support for our hypothesis as well as recent published literature[24] led us to explore whether life course decisions and behaviors of the mother, which precede the child's birth, may help to explain test score gains among children whose mothers were exposed to the cash transfer for longer periods of time. A mother's decision to, for instance, pursue higher education, marry, delay fertility, or refrain from smoking during pregnancy all could plausibly lead to improvements in child's test scores. Results from the exploration (Model 3, Tables 2 and 3) indicate that several of these variables predict children's test scores. Inclusion of these variables, moreover, attenuates the interaction term by ~20%. The interaction term, however, reaches conventional levels of statistical detection for both math and reading, which indicates that these factors may not fully account for American Indian children's gain in test scores.

## Sensitivity analyses

We conducted several additional checks to assess robustness of results. First, to support the validity of the DiD model, we tested the parallel trends assumption in the pre-treatment period[21] by interacting a time-invariant treatment indicator (American Indian status) with the age of the mother in 1996 minus 18 years of age, and then testing whether the coefficient of the interaction term (i.e., American Indian*pre_treatment) rejects the null for the periods prior to treatment. Results of the American Indian*pre_treatment coefficient in the pre-treatment period do not reject the null for either math or reading test scores (see Supplementary Tables 2 and 3), which supports parallel trends in the pre-treatment period.

Second, we restricted the analysis to mothers (G2) who received between 0 and 12 years of duration exposure by 1996 to rule out the possibility that outliers in exposure drive results. Inference for both math and reading did not change (Supplementary Tables 4 and 5). Third, we restricted the mother's (G2) age of delivering children to 16–35 years. We arrived at this range by inspecting the age distribution of mothers at the time of the child's (G3's) birth, by American Indian status, and dropping the maternal ages for which fewer than 10 participants fell into that cell. This sensitivity check rules out the possibility that high "outliers" in maternal age drive results. Findings remain similar to those in columns 2 of Tables 2 and 3, albeit with less precision owing to dropping 8% (math) and 11% (reading) of observations after these restrictions (Supplementary Tables 6 and 7). Fourth, to rule out the possibility that trends over time in test scores (such as declines reported nationally[22] and in rural areas[25]) drive results, we controlled for test year in several ways (including a continuous year variable and, separately, test year indicator variables) and re-ran analyses. This time control also adjusts for any potential response to the 2007-2009 Great Recession. Inference for the American Indian*duration coefficient does not change (Supplementary Tables 8–11).

## Discussion

We investigated whether childhood investments, in the form of family cash transfers, could improve human capital outcomes in the next generation of children. We focused on a Southeastern American Indian tribe in rural North Carolina who, via a natural experiment by the

**Table 1 | Maternal and Birth Characteristics of American Indian and non-American Indian children in Jackson, Swain, and Graham counties whose birth record linked to third grade test scores from 2008 to 2017**

|  | AI | | Non-AI | |
| --- | --- | --- | --- | --- |
|  | N | %[a] | N | %[a] |
| Maternal age (yrs) |  |  |  |  |
| 17 or younger | 66 | 8.9 | 138 | 3.9 |
| 18–24 | 407 | 55.0 | 1447 | 40.8 |
| 25–29 | 168 | 22.7 | 969 | 27.3 |
| 30 or older | 99 | 13.4 | 995 | 28.0 |
| Maternal educational attainment |  |  |  |  |
| Less than high school | 251 | 33.9 | 675 | 19.0 |
| High school graduate | 295 | 39.9 | 1158 | 32.6 |
| >High school | 193 | 26.1 | 1713 | 48.3 |
| Married |  |  |  |  |
| Yes | 279 | 37.7 | 2629 | 74.1 |
| No | 461 | 62.3 | 919 | 25.9 |
| Used tobacco during pregnancy | 173 | 23.4 | 753 | 21.2 |
| Infant sex |  |  |  |  |
| Male | 382 | 51.6 | 1815 | 51.1 |
| Female | 358 | 48.4 | 1734 | 48.9 |
| Low weight birth (<2500 gm) | 27 | 3.6 | 217 | 6.1 |
| Preterm (<37 weeks) | 48 | 6.5 | 299 | 8.4 |

[a]Column percents may not sum to 100 due to missing values for that variable.

**Table 2 | DiD regression results predicting third grade math Z-score for 4289 children in Jackson, Swain, and Graham Counties, 2008–2017, as a function of American Indian race/ethnicity, duration of mother's exposure to family cash transfer as a child, and other covariates**

| Variables | Model 1 | | | Model 2 | | | Model 3 | | |
| --- | --- | --- | --- | --- | --- | --- | --- | --- | --- |
|  | coef | p-value | 95% CI | Coef | p-value | 95% CI | Coef | p-value | 95% CI |
| AI race/ethnicity (AI) (reference = non-AI) | −0.466 | <0.0001 | [−0.568 −0.364] | −0.420 | <0.0001 | [−0.520 −0.320] | −0.315 | <0.0001 | [−0.415 −0.214] |
| Duration of potential exposure of mother to family cash transfer in childhood (duration)[a] | −0.042 | <0.0001 | [−0.051 −0.033] | −0.041 | <0.0001 | [−0.050 −0.032] | −0.014 | 0.030 | [−0.026 −0.001] |
| Interaction of AI * duration | 0.029 | 0.006 | [0.009 0.050] | 0.025 | 0.015 | [0.005 0.045] | 0.020 | 0.045 | [0.001 0.040] |
| Offspring Age at test (years) | – | – | | 41.769 | 0.021 | [6.428 77.110] | 32.289 | 0.065 | [−2.062 66.64] |
| Offspring Age at test$^2$ (years) | – | – | | −4.076 | 0.030 | [−7.760 −0.392] | −3.136 | 0.086 | [−6.716 0.444] |
| Offspring Age at test$^3$ (years) | | | | 0.130 | 0.046 | [0.002 0.258] | 0.100 | 0.116 | [−0.025 0.224] |
| Male sex at birth (reference = female) | – | – | | 0.026 | 0.350 | [−0.029 0.081] | 0.017 | 0.534 | [−0.037 0.071] |
| Maternal age (years) | | | | | | | −0.001 | 0.882 | [−0.007 0.006] |
| Married (ref: not married) | – | – | – | – | – | – | 0.103 | 0.002 | [0.039 0.167] |
| Mother's terminal degree is HS (ref: <HS grad) | – | – | – | – | – | – | 0.139 | 0.0003 | [0.064 0.215] |
| Mother pursued >HS degree (ref: <HS grad) | – | – | – | – | – | – | 0.487 | <0.0001 | [0.406 0.569] |
| Tobacco use during pregnancy | | | | | | | −0.119 | 0.0006 | [−0.187 −0.051] |
| Infant Birth Weight (continuous, in grams) | – | – | | | | | 0.0001 | 0.019 | [0.0000 0.0001] |
| Intercept included | yes | | | yes | | | yes | | |

[a]This variable indicates the number of years of possible receipt of cash transfer for AI participants; for non-AI participants, it represents an age control.
[b]We applied generalized estimating equation regressions[50] using maximum likelihood estimators, to predict the test score outcomes (PROC GENMOD in SAS). We used two-tailed tests for all statistical analyses.

**Table 3 | DiD regression results predicting third grade reading Z-score for 4254 children in Jackson, Swain, and Graham Counties, 2008–2017, as a function of American Indian race/ethnicity, duration of mother's exposure to family cash transfer as a child, and other covariates**

| Variables | Model 1 | | | Model 2 | | | Model 3 | | |
|---|---|---|---|---|---|---|---|---|---|
| | Coef | p-value | 95% CI | Coef | p-value | 95% CI | Coef | p-value | 95% CI |
| AI race/ethnicity (AI) (reference = non-AI) | −0.546 | <0.0001 | [−0.649 −0.442] | −0.494 | <0.0001 | [−0.595 −0.392] | −0.406 | <0.0001 | [−0.415 −0.214] |
| Duration of potential exposure of mother to family cash transfer in childhood (duration)[a] | −0.045 | <0.0001 | [−0.054 −0.036] | −0.043 | <0.0001 | [−0.052 −0.034] | −0.012 | 0.063 | [−0.026 −0.001] |
| Interaction of AI * duration | 0.034 | 0.001 | [0.013 0.055] | 0.028 | 0.007 | [0.008 0.049] | 0.024 | 0.018 | [0.001 0.040] |
| Offspring Age at test (years) | – | – | – | 77.452 | <0.0001 | [41.111 113.793] | 66.880 | 0.0002 | [−2.062 66.64] |
| Offspring Age at test[2] (years) | – | – | – | −7.748 | <0.0001 | [−11.539 −3.958] | −6.697 | 0.0004 | [−6.716 0.444] |
| Offspring Age at test[3] (years) | – | – | – | 0.256 | 0.0001 | [0.124 0.387] | 0.222 | 0.0007 | [−0.025 0.224] |
| Male sex at birth | | | | | | | | | |
| (reference = female) | – | – | – | −0.105 | 0.0002 | [−0.160 −0.050] | −0.109 | <0.0001 | [−0.037 0.071] |
| Maternal age (years) | | | | | | | 0.004 | 0.208 | [−0.007 0.006] |
| Married (ref: not married) | | | – | – | – | – | 0.003 | 0.930 | [0.039 0.167] |
| Mother's terminal degree is HS (ref: <HS grad) | | | – | – | – | – | 0.182 | <0.0001 | [0.064 0.215] |
| Mother pursued >HS degree (ref: <HS grad) | | | – | – | – | – | 0.566 | <0.0001 | [0.406 0.569] |
| Tobacco use during pregnancy | | | | | | | −0.056 | 0.111 | [−0.187 −0.051] |
| Infant birth weight (continuous, in grams) | | | – | – | – | – | 0.0001 | 0.016 | [0.0000 0.0001] |
| Intercept included | Yes | | | Yes | | | Yes | | |

[a]This variable indicates number of years of possible receipt of cash transfer for AI participants; for non-AI participants, it represents an age control.
[b]We applied generalized estimating equation regressions[50] using maximum likelihood estimators, to predict the test score outcomes (PROC GENMOD in SAS). We used two-tailed tests for all statistical analyses.

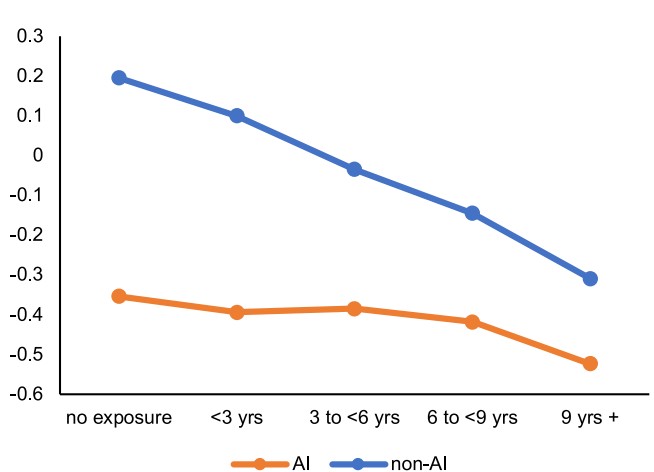

**Fig. 3 | Third grade math Z-Score[†] (fitted regression values from Table 2, Model 2) by American Indian status, by duration of exposure to payments before 18 years.** Within the context of the declining trend in third grade math test scores in this rural population (which mirrors national trends), the race-based disparity in test scores narrows for American Indian children whose mothers had a relatively greater duration exposure. American Indian scores are represented by the orange line. Non-American Indian scores are represented by the blue line.

**Fig. 4 | Third grade reading Z-Score[†] (fitted regression values from Table 3, Model 2) by American Indian status, by duration of exposure to payments before 18 years.** Within the context of the declining trend in third grade reading test scores in this rural population (which mirrors national trends), the race-based disparity in test scores narrows for American Indian children whose mothers had a relatively greater duration exposure. American Indian scores are represented by the orange line. Non- American Indian scores are represented by the blue line.

introduction of a successful casino, received a large cash transfer. Findings indicate statistically significant increases in both reading and math third grade test scores among students born to American Indian mothers with more years of exposure to the cash transfers as children. Results, which control for general changes in the region over time that could have benefited American Indian and non- American Indian students equally, support the hypothesis that large early-life investments show human capital benefits into subsequent generations.

Many American Indian (G2) mothers who were very young in 1996 (i.e., <5 years old) have children that are scheduled to attend third grade after 2017—the last year in which we could link test score information. This circumstance means that our analysis includes very few (G2) mothers who had early-life exposure (i.e., from infancy to age 5) to the cash transfer. Our results may therefore underestimate the potentially larger benefit of cash transfers (especially before age 5 years among G2) that may accrue to the subsequent generation of

American Indian children and produce large returns to health and education[26,27].

The magnitude of the statistically significant test score increases in reading and math for children born to American Indian mothers seems reasonable in relation to prior interventions in North Carolina[23]. The slightly larger benefits observed for reading, moreover, cohere with the notion that non-school factors play a substantial role. The education literature generally finds that reading skills develop in much broader (i.e., non-school) settings relative to math skills[28–30]. This work implies that our discovered results likely do not arise from unmeasured factors in which American Indian mothers (but not non-American Indian mothers) chose high-performing schools for their children. We also note, importantly, that non-American Indian children show a steep declining trend over time in test scores, and that American Indian children do not show increases in the absolute level of test scores with increased exposure to the cash transfer. National studies similarly find declining trends in test scores over this time period[22], as well as persistently lower test scores among white and American Indian children in rural areas[25,31] of the US (vs. suburban and urban areas). Explanations for these geographic patterns and time-trends remain elusive. We encourage more careful research in this area to understand the broader national educational landscape within which the cash transfer accrues to American Indian families and children in this rural population.

Since the introduction of the casino in the late 1990s, the Tribe constructed several new facilities including healthcare centers and educational academies. The New Kituwah Academy[32], for instance, is a private facility (accredited in 2015) which offers, among other programs, dual-immersion elementary school education focused on preserving the Cherokee language, culture, traditions, and history. Whereas American Indian children enrolled in this Academy would not appear in our dataset (i.e., NCERDC linked test scores only for public school-enrolled children), this resource as well as others may benefit human capital especially among American Indian children. Although we have no reason to believe that these benefits covary with the number of childhood years remaining at the start of the family cash transfer, our methods cannot rule out this explanation. We, however, note that much of the infrastructure improvements on Tribal lands remain available to all residents regardless of race/ethnicity. Therefore, our DiD analyses help to control for this rival explanation.

Whereas our findings are among the first to document statistically significant intergenerational test score improvements—25 years after the inception of large family cash transfers—several caveats deserve mention. First, the magnitude of the gains to American Indian children depicts a narrowing of the differences between American Indian and non-American Indian math and reading scores since the onset of the cash transfer in 1996. Despite the higher American Indian math and reading scores, the large American Indian/non-American Indian score gap in math (0.46 SD) and reading (0.54 SD) scores did not close during this time. The latter is as expected considering cash transfers alone are unlikely to rectify the education effects of multi-generational discrimination among American Indian and non-American Indian populations[33]. This discrimination includes past and present unequal treatment as well as structural factors that may lead to a higher prevalence of predictors of low educational attainment among American Indian populations (e.g., poverty, residing near low quality schools, high levels of teen pregnancy; see Demmert and colleagues)[30,34].

Second, NCERDC could not link the full population of births in this region to their third-grade test score. Non-matches are attributed to moves out of state, private school attendance, name changes, or errors in spelling on records. Third, substantial missing/unknown paternity on the birth file did not permit an examination of whether having an American Indian father who received the cash transfer, or having two American Indian parents (vs. solely an American Indian mother) that received the cash transfer, confers stronger intergenerational

associations. Fourth, given the nature of the timing of cash transfers to this population, we cannot determine which factor (child age at initiation of cash transfer or duration of cash transfer exposure before 18 years) seems most relevant in designing new interventions. Fifth, some other work examining this large cash transfer to this population shows adverse outcomes, such as risk of accidental death during months of large casino payments[35,36]. This circumstance indicates that any policy discussion about the value of family cash transfers to the next generation should include a careful assessment of their costs and benefits to all generations as well as an assessment of the type (e.g., in-kind vs. cash) and frequency (e.g., lump sum or monthly payment) of the transfer.

Whereas the population-based nature of our linked datasets provides a larger sample size than do cohort studies of this population (i.e., GSMS), the birth and test score data lack contextual variables that may illuminate mechanistic pathways. American Indian mothers with more years of exposure to the family cash transfer as children could, for instance, make a variety of life course decisions that ultimately benefit their children. Previous work on this population finds that fertility[37], attitudes around fertility timing[38] as well as educational attainment[13] may change after the introduction of the family cash transfer. Recent work also finds that American Indian mothers exposed for a longer duration to the cash transfer show improved maternal/infant health at birth[24]. These pathways, as well as prenatal investments or changes in parenting quality, could account for gains in children's test scores. We await the availability of additional contextual data, as well as a richer set of school-level variables (e.g., attendance, test scores at later ages) in coming years.

Within the context of the secular decline in third grade test scores in this rural population (Supplementary Figs. 1 and 2), the American Indian / non-American Indian disparity in test scores narrows as mothers of American Indian children have a relatively greater duration of exposure. Whereas we infer that this finding arises from the benefit of the cash transfer to American Indian families, we cannot rule out the possibility of unmeasured confounding. Such a confounder would have to correlate positively with our exposure (but not be caused by it), occur only among American Indian families (but not among non-American Indian families), and vary positively with third grade test scores. School-based investments particular to American Indian children that concentrate in recent years, or broader employment gains to American Indian families that concentrate in recent years, could meet these criteria. We, however, know of no such trend in school-based investments unique to American Indian children in public schools. In addition, both American Indian and non-American Indian adults show employment gains following the opening of the casino, which minimizes the plausibility that this factor introduces bias.

The casino opening led to several community improvements besides the cash transfer to tribal members. The tribe designated half of the gaming revenues to community investments, including behavioral health, drug abuse prevention, health care, education, and social services[39,40]. In addition, the casino itself is the largest employer in the region and boosts other local businesses[41]. These improvements may lead to gains in health and functioning for all American Indian members (regardless of age) as well as non-American Indian individuals in the study region.

The establishment of the cash transfer payments among this population in the 1990s substantially raised median income in a community that previously exhibited a high poverty rate. Between the years of 1995 and 2000, the percent of American Indian families below the poverty line fell from almost 60% to less than 25%[42]. This circumstance, coupled with accumulating literature documenting improved adult health among recipients who were at earlier childhood ages at the onset of the family cash transfer[17], compelled us to examine the potential intergenerational benefits among those who were young in 1995 and later decided to have children. An intuitive follow-up question involves whether these intergenerational associations would

persist, or even become stronger, among those who were in infancy or under age five at the inception of the family cash transfer in the 1990s and later had children of their own. For American Indian females born in this region in 1995, we can expect their children to have completed third grade and test scores to be available by 2050. In the near term, however, we encourage replication in other settings in the US to determine external validity. A more complete picture of educational outcomes (e.g., subject-matter test scores other than reading and math, school attendance, social and emotional well-being), which we aim to collect in future work, may also better capture academic ability. Other extensions of this work should identify potential pathways in which less impoverished childhood environments affect later-life adult school choice, fertility decisions, and parental investments that in turn enhance human capital of the next generation.

## Methods

### Study population

We examined American Indians in Jackson, Swain, and Graham counties in North Carolina as a proxy for the Eastern Band of Cherokee. According to the 2020 Census, American Indian residents comprise 14.8% of the population in these three counties. No other federally recognized, state recognized, or even unrecognized Tribes claim lands in the western North Carolina area, and the Eastern Band of Cherokee have historically been the only Tribe in this region of western North Carolina. Previous studies have used the census indicator of American Indians as a proxy for Eastern Band of Cherokee in this region[42]. These American Indians residents received the large family cash transfer beginning in 1996. By contrast, non- American Indians residents in these counties received no cash transfer but (as with the American Indian population) experienced the broader economic and infrastructural changes to that region. We therefore use children born to non- American Indians residents of Jackson, Swain, and Graham counties as a comparison group when examining the relation between the family cash transfer and educational outcomes among American Indians residents' children.

### Inclusion and ethics statement

This study was completed using education and birth records from a number of counties in western North Carolina. The data for the current manuscript were obtained from the North Carolina Education Research Data Center (NCERDC), which houses data files from State of North Carolina administrative records[43]. Through data use agreements between Duke University and the State of North Carolina, the NCERDC receives state data files with identified records, merges files as needed, de-identifies the merged files, and then provides access to de-identified files to researchers. None of the NCERDC staff members who worked on the current data set are researchers or authors of the current study. The NCERDC is described here: https://childandfamilypolicy.duke.edu/north-carolina-education-research-data/. This study is relevant to the educational functioning of families receiving the cash transfer in western North Carolina, but this was not determined in collaboration with local partners. The roles and responsibilities for compiling the data were agreed upon by collaborators ahead of time.

This study was approved by the IRB at Duke University which is located in North Carolina but not specifically in western North Carolina. Also, the research does not result in discrimination as it was focused on a quasi-experiment design resulting from the introduction of a community-wide transfer. The Southeastern American Indian Tribe which co-generated (along with the casino) the cash transfer has promoted this transfer as a public good. We have taken local and regional research relevant to our study into account by citing prior studies of this cash transfer.

### Variables and data

Starting in the third grade, North Carolina conducts end-of-grade standards-based achievement tests for math and reading for all students enrolled in public school. The reading and mathematics tests align with the North Carolina Standard Course of Study[44]. We used third grade test scores as our key dependent variable because education scholars view these measures as a stable indicator of student achievement and a reliable predictor of longer-term educational outcomes, not only nationally but also in North Carolina[23,45]. Test scores by third grade predict both likelihood of high school graduation and college attendance[19,20]. We standardized each raw score to Z-score values using the mean and standard deviation (SD) of all third-grade scores in North Carolina for that test year. This standardization permits direct comparison of student scores across years because it controls for variation over time in difficulty or scaling of the state tests (e.g., if mean test scores show a trend over time, the Z-score values [normed within each test year] are less subject to such trends).

We acquired third grade math and reading test scores among infants born in Jackson, Swain, and Graham counties using linked administrative data files from the Duke University North Carolina Education Research Data Center (NCERDC). The NCERDC receives educational administrative data files from the North Carolina Department of Public Instruction (NC DPI), which collects files submitted annually by each of 115 school districts. To identify the child's county of birth, NCERDC links individual birth records from the Birth File of the North Carolina Office of Vital Records for all children born in the state with education records from NC DPI. The sample includes only children born in North Carolina and then enrolled (by third grade) in a public elementary school in the state. This process necessarily excludes children who enroll in a private school as well as those whose families moved out of North Carolina by third grade. Over 200 peer-reviewed publications use NCERDC-linked data, which attests to the quality and coverage of the dataset[46].

Beginning in 2008, in our study region NCERDC reports a match rate of >74% between birth records and third grade test scores. 2017 represents the last year for which we have matched data available at the time of our study. Our test population includes over 4000 American Indian and non- American Indian children who have a valid third grade test score from 2008 to 2017−and who were born from 1998 to 2009 in Jackson, Swain, and Graham counties.

Prior literature finds a positive relation between American Indian later-life health and the number of years during which the individual was exposed to the family cash transfer before reaching age 18 years[17]. This relation coheres with the notion that the duration of the family cash transfer during childhood can exert a positive influence later in life. We, similarly, reasoned that additional benefits could include life-course maternal investments and behaviors which in turn may improve the next generation's educational outcomes. For this reason, and consistent with prior literature[17,42], we used as the primary exposure the number of years before age 18 that the index individual's family received the cash transfer.

The Birth File contains several variables that may control for confounding bias but do not plausibly lie on the causal pathway between family cash transfer and the next generation of children's test scores. These variables, which show associations with test scores, include infant sex and child age (i.e., date of birth). We retrieved these variables from the birth file and used them (as well as other variables in the birth file [maternal education, maternal age], described below in Analysis section) as controls for potential confounding. We determined infant sex based on sex assigned at birth, as recorded on the birth certificate.

### Analysis

All data analyses were conducted using SAS version 9.4. Examination of American Indian and non-American Indian cohorts at varying ages at the inception of the family cash transfer in 1996 confers the methodological benefit of using the family cash transfer as a "natural experiment" which randomly assigns income to American Indian

families. We employ a "difference-in-difference" (DiD) regression strategy to isolate potential benefits of the family cash transfer on educational outcomes of children born to American Indian mothers who were relatively young in 1996—the first year of the family cash transfer program. This approach uses a series of control populations to adjust for unmeasured confounding and other threats to validity. It remains plausible, for instance, that the level of social, educational, and economic resources increased over time in Jackson, Swain, and Graham counties in ways that benefited younger-age cohorts in 1996 (relative to older-age cohorts in 1996). This circumstance could result in improved math and reading test scores of children born to younger (relative to older) cohorts. Such a circumstance would confound our test if we falsely attributed this positive relation to the duration of the family cash transfer in childhood.

Our DiD regression approach minimizes the problem of unmeasured confounding. This strategy compares the test scores outcome of children born to American Indian mothers who were young in 1996 to that of children born to American Indian mothers who were relatively older in 1996. Importantly, we also adjust for general cohort differences in access to social, educational, and economic resources in Jackson, Swain, and Graham counties.

The key features of a DiD design involve (i.) comparison of outcomes between two alternative treatment regimes (i.e., treatment and control), (ii.) the availability of pre-treatment and post-treatment time periods in both the treatment and control group, and (iii.) a well-defined study population[21]. We augment this standard DiD with a time-varying treatment effects design, also called DiD with treatment as intensity of exposure. This design assumes that the relation of the treatment to the outcome increases with longer duration of exposure to the treatment. In our case, duration of exposure to the cash transfer as a child serves as the intensity of exposure for American Indian mothers.

The DiD approach (shown below) minimizes the problem of unmeasured confounding. This strategy compares the test scores outcome ($\theta$, representing third grade math or reading Z-score) of children born to American Indian mothers who were young in 1996 to that of children born to American Indian mothers who were relatively older in 1996. Importantly, we also adjust for general cohort differences in access to social, educational, and economic resources in Jackson, Swain, and Graham counties by subtracting the difference in test scores observed between children born to non-American Indian mothers who were young in 1996 and non-American Indian mothers who were relatively older (i.e., around age 17) in 1996.

$$\left[ \left( \theta_{mother\ younger\ in1996} - \theta_{mother\ older\ in1996} \right) i_{AI\ residents} \right.$$
$$\left. - \left( \theta_{mother\ younger\ in1996} - \theta_{mother\ older\ in1996} \right) i_{non\ AI\ residents} \right] \quad (1)$$

Social scientists have employed this approach to examine the effect of large "shocks" on children's outcomes[28,47–49].

Estimation of the equation above entails pooling data for American Indian and non-American Indian births in Jackson, Swain, and Graham counties, and regressing the third grade test score outcomes from 2008 to 2017 (Z-score for math, and Z-score for reading) on a dichotomous indicator capturing (1) American Indian race/ethnicity (as measured by mother's race/ethnicity from the Birth file), (2) a continuous indicator of childhood years remaining before age 18 at the start of the family cash transfer and the two-way interaction between American Indian race/ethnicity and childhood years remaining at the start of the family cash transfer. The estimate of interest is the coefficient on the two-way interaction term, which captures the difference in test score outcome between American Indian children born to residents who were relatively young in 1996 and those who were older in 1996, net of that same difference in

non-American Indian children. Specifically, we examine the influence of the cash transfer by regressing Y (children's test scores) on $X_1$ (time exposed before age 18; continuous) and $X_2$ (American Indian status), and then testing whether the relation between Y and $X_1$ differs by American Indian Status ($X_2$). The DiD regression also includes controls for the child's age in months at third grade test and assigned sex at birth. We applied generalized estimating equation regressions[50] using maximum likelihood estimators to predict the two continuous outcomes (PROC GENMOD in SAS). The test score data (for both math and reading) meet the assumptions for use of these methods. We used two-tailed tests for all statistical analyses.

If we discovered support for a positive relation between the interaction term and Z-score of test (i.e., more childhood years remaining at the start of family cash transfer varies positively with subsequent generation's third grade test score), we then explored potential pathways of this association. Such an exploration included the addition of maternal education, maternal behavior during pregnancy, and infant health information contained in the Birth File. In addition, as a falsification check we examined the assumption of parallel trends in a DiD framework by testing pre-treatment trends between the treated group and the control group prior to the treatment. To do so, we interacted a time-invariant treatment indicator (American Indian status) with the age of the mother in 1996 minus 18 years of age—but only among mothers 18 years or older in 1996 and therefore never exposed as a child to the cash transfer treatment—and then tested whether the coefficient of this interaction term (American Indian*pre-treatment) rejected the null for both children's math and reading test scores (see Supplementary Tables 2 and 3). Failure to reject the null would satisfy the parallel trends assumption in the pre-treatment period.

### Reporting summary
Further information on research design is available in the Nature Portfolio Reporting Summary linked to this article.

## Data availability
The individual-level linked birth records and education outcomes, deriving from existing administrative records, are housed by the NCERDC and derive from existing administrative records. The individual-level raw data are available under restricted access given the usage of personal identifiable information, the state of North Carolina's restrictions on dissemination without prior consent, and the regulations set by the IRB protocol (Protocol: Pro00090215 with Duke University). The raw data are protected and are not available due to privacy laws. Request for raw data can be made to the NCERDC here: https://childandfamilypolicy.duke.edu/north-carolina-education-research-data/. Data are only provided to researchers who meet the requirements of the NCERDC Data Use Agreement which stipulates primary affiliation with an institution of higher education, non-profit organization, or government agency within the United States. Additional information can be found at the link provided above. In addition, to comply with open science requirements and that of NCERDC, the processed group-level data used in this study are available within the Figshare database[51] and are available here: https://doi.org/10.6084/m9.figshare.26288080.v1. These data include the covariance matrix of the data analyzed along with a vector of means, standard deviations, and number of observations, separately by American Indian and non-American Indian participants. This information allows interested readers to re-create the regression analyses. The data file also provides the summary data points used to create all figures.

## Code availability
The SAS program code is available upon request to the first Author, who can provide the code via email.

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

## Acknowledgements

This work was supported by the Eunice Kennedy Shriver National Institute of Child Health and Human Development (5R01HD093651-05)(K.A.D.).

## Author contributions

T.A.B. contributed to the conceptualization, methodology and formal analysis. B.B. contributed to the formal analysis and visualization. K.A.D. and J.E.L. contributed to the acquisition of data. C.L.O. and W.E.C. contributed to the interpretation of data. All authors contributed to writing, reviewing, and editing of the manuscript.

## Competing interests

The authors declare no competing interests.
