## [Peer Review File · Nature Communications]

Reviewers' Comments:

Reviewer #1:

Remarks to the Author:

This paper reports the intergenerational effects on educational attainment (reading and maths scores) of a cash transfer natural experiment. This is really important work that should inform government programmes. The earlier papers on the first generation effects of this programme (a natural experiment due to casino royalties) have been very influential, and this paper, which goes into the outcomes of the next generation, adds substantively to what was known. I thus recommend publication, and find the analysis, data and reporting to be of good quality.

I have a few comments, which are quite minor.

1. There is no visualization of the results, which is a shame. It seems to me it ought to be quite possible to make some graphs, either from the model predictions or directly from the data, that show how average test score increases in years' exposure for the American Indian but not the other families. This would also allow an intuitive appreciation of the effect size relative to e.g. the AI-nonAI difference, or the secular trend.

2. I found the explanation of the difference in difference regression (lines 146 and also 360) a bit obscure. In fact, it seems to me, this is just a regression model which enters years' exposure, ethnicity and their interaction, with the interaction being the critical test. Or am I missing something? I can't see why this needs to be described as d-in-d at all. I usually associate that term with longitudinal studies where you explain the temporal change in the outcome using temporal change in the predictor. But this study has a one-off outcome and that terms does not apply in quite the same way. In any event, a more intuitive explanation of the modelling strategy is needed (and this would couple well with a figure, see previous point).

3. (Point coming from an Open Science bore). I understand why the authors can only release a de-identified dataset. But rather than saying 'on reasonable request', could this not simply be placed in a repository? 'On request' data sharing is frowned on in my discipline these days because so many authors don't respond to requests, move on, etc.

4. In the income and health literature we often find that estimates from interventions are rather different from the estimates you get from observational studies. Though the authors mention that the effects here were about the magnitude you would expect from a government programme in North Carolina, is there more that could be said. For example, you could work out a prior expectation from the observational relationships between childhood income and parental human capital, and then parental human capital and offspring human capital. Or, do we know anything about the educational attainment and income of G2, so we could work out to what extent G3 have 'regressed to the mean', or magnified the different made by the cash transfer in G2. So, all in all, it would be good to situate the effect size relative to what was already known from other designs of study.

Reviewer #2:

Remarks to the Author:

This article examines effects of casino-related cash gifts on the intergenerational transmission of academic performance in a Native American community. The paper has several strengths, including the multigenerational design, quasi-experimental income manipulation, sample size, cognitive tests, and difference-in-difference statistical models comparing to both non-AI children as well as children born to AI mothers who were relatively older. This is a valuable paper since there is so little multi-generational research on this topic. A few minor comments:

On pg 14 the authors note: "First, the magnitude of the gains to AI children narrows, but does not close, the large AI / non-AI score gap in math (0.46 SD) and reading (0.54 SD)." I think this finding deserves some more thought/speculation, e.g. educational effects of multi-generational discrimination and oppression experienced by AI communities are unlikely to be fully ameliorated by cash gifts.

Throughout, it would help readers if there was more detail on what was considered "younger", "older", "longer duration" or note that these are continuous time variables including the range of eg cash transfer duration without having to refer to the Methods section.

Figure 2 compares reading/math between AI to non-AI groups, but that is not quite the main highlighted finding. Perhaps add a panel that describes the cash duration effect on reading and math.

Reviewer #3:

Remarks to the Author:

Thank you for the opportunity to review this interesting article examining 2nd generation effects of a natural experiment involving a cash transfer related to the initiation of a casino for American Indian families. The study represents an important addition to the literature regarding the potential benefits of cash transfers, highlighting that potentially particularly for lower resource families the additional cash resources may yield long-term benefits. While there are some limitations to the study, as acknowledged by the authors (in particular the lack of information about fathers exposure to the cash transfers), on balance the strengths outweigh the weaknesses. Major:

1. The ages of exposure to cash transfer is hard to clearly understand. The abstract uses the phrase: "during the sensitive period of childhood," though given that childhood may include 0-18 years, that is perhaps an unusually broad set of years to all be classified as "a sensitive period." Given that past work (e.g., Chetty's moving to opportunity analyses) can use quite broad age categories, I wish for greater discussion and the potential for analyses to try to disentangle questions related to the duration of the exposure and the age of exposure (given the possibility that increased plasticity/responsibility may occur at younger ages or alternatively during puberty, specifically).

Minor:

2. Figure 2. It would be helpful to include a color key on the actual figure. Also, consistent font choice and use of black font would enhance readability. I suggest capitalizing "third" for consistency with the other words in the title.

3. Use of a serial comma throughout would enhance clarity

4. I defer to the journal editor regarding whether the written labels in Figure 1 conform to style guides

5. There is inconsistency in whether a space is used (and on both sides) with equal and less than signs

Reviewer #4:

Remarks to the Author:

This paper attempts to identify the impact of receiving a cash infusion for a longer (vs shorter) period of time during childhood on offspring academic test scores, attempting to leverage a large, positive economic shock that was created by the tribal ownership of a casino among the Eastern Band of Cherokee Indians. Payments to tribal members began in 1996 and were issued to all tribal members on a yearly basis thereafter. These payments were about \$5000 a year. The casino and these payments substantially alleviated poverty levels for the tribe. The results could be quite noteworthy as it is difficult to isolate the impacts of additional income, especially income not tied

to work. However, the paper currently lacks sufficient detail and robustness checks to bolster confidence in the results and would be substantially improved by providing additional analysis and/or thought around how any unmeasured or collinear biasing factors might impact the results.

While difference-in-difference (DID) designs are very common for evaluating policies or “shocks”, the particular DID approach taken in this paper is a fairly atypical difference-in-difference strategy. Typically, a DID study would compare the change in the outcome for those ‘exposed’ to a shock or policy from before to after “treatment”/shock to the change in an unexposed group over the same period of time. In this paper, the authors are comparing differences in outcomes for four different groups, two of whom have differing durations of exposure to cash payments, and two of whom were never exposed. In essence, we do not observe the “treated” group before “treatment”; we only observe people who have received treatment beginning earlier vs later in childhood. The identification of the impact of cash for a longer period during childhood relies on the difference in outcomes in 2008-2017 for “younger in 1996” vs “older in 1996” among non-American Indians being a good counterfactual for the difference between “younger in 1996” vs “older in 1996” among American Indians in 2008-2017. Adding some sensitivity or falsification tests seems necessary to bolster the analysis and conclusions. I’m not sure exactly what these could be, partially due to the fact that it has been a very long time since the cash payments started, so it would be difficult to assess pre-policy trends among the different populations. But if the authors could assess an outcome that would conceivably not be responsive to receiving cash for a longer duration of childhood, that would bolster the results. Alternatively, just helping the reader think through what type of bias would need to be operating to invalidate the results would also be helpful.

As above, the main strategy for assessing impacts of receiving cash payments for a longer (vs shorter) duration is to compare academic outcomes during 2008-2017 for American Indian children of birthing people who would have received cash payments for a longer duration by virtue of being relatively younger in 1996 when the payments began to outcomes for American Indian Children born to birthing people who were older in 1996 and then additionally compare the difference in outcomes for these two populations to the difference in outcomes for non-American Indian birthing people who were younger in 1996 to non-American Indian people who were older in 1996. Because the identification is linked to being older or younger in one calendar year, the issue of time in all its forms (age, year, cohort) seem particularly important to pay attention to, even if just by explaining to the reader what assumptions are being relied on. Currently, the paper does not do enough to subject the results to robustness checks and falsification tests that would help the reader feel more confident that the results are indeed due to a longer duration of casino payments during the childhood. Below I suggest some additional information that would help probe the results and assess the degree to which they may or may not be due to other factors.

More information is required to explain the age distribution of the sample and overlap in maternal age for people exposed at younger versus older ages to the beginning of the cash transfers. A table of how many AI and non-AI are in different ‘bins’ of duration for each year of the outcome data would be very helpful for visualizing overlap. I think the analysis should be limited to women with maternal ages for which there is good overlap for those who received the payments for a longer duration during childhood versus a shorter duration.

From my calculations, it seems like there would actually be very few people in the sample who would be under 5 when cash payments began and have children in third grade between 2008 to 2017. Specifically, birthing people who are 5 in 1996 will be 26 in 2017. If third graders are approximately 8 years old, they would have had children at 18, which is not highly unusual, but would be a relatively small portion of the sample. Seems that even fewer would have children before that. This has at least 2 implications: This means that effects we are seeing are due to receiving cash for a longer period in childhood, but still largely outside of the birth to five range in which previous literature has shown that investments produce large returns to health and educational outcomes.

Second, people who were younger vs older in 1996 are also going to be different on either

maternal age at birth or the year their children are in third grade. I do not think there is a way to disentangle these potentially different effects, both of which could impact test scores. Someone who was 5 in 1996 and had a child at 18 would have third grade test scores for their children in 2017, whereas someone who was 10 in 1996 and had a child at 18 would have third grade test scores in 2012. There could be secular trends in test scores over this period and the assumption of the models are that these are the same for AI versus non-AI children. While some aspects may not be directly testable, if the authors discuss these limitations and assumptions and are able to reason that, if anything, we might expect effects in the opposite direction than what we are seeing, that could also bolster the arguments.

Additionally, I think that people who were younger (vs older) in 1996 would have received the cash payments for a longer total period of time, not just in childhood (although I'm not completely sure about this; this might be taken care of through the wide interval of outcome data collection 2008-2017).

In a related comment, the time period for which outcomes are compared is 2008 to 2017 is a pretty long period of time. Are there trends in test scores over this period, are the trends parallel for AI and non-AI. It seems like the identification strategy needs to have all 4 groups – younger in 1996 AI, older in 1996 AI, younger in 1996 non-AI, older in 1996 non-AI in each or most of these outcome years. If you do, than it would seem you could include year fixed effects or at least year trends. However, including year fixed effects would, I think by structure, mean that you are comparing people of different maternal ages, so limiting to a subset of maternal ages might be necessary, at least as a sensitivity analysis.

More details of the calculation of duration are needed. Specifically, although it is implied through the naming of the variable as 'duration', it should be clearly stated whether each participant's age in 1996 is subtracted from 18 or the other way around. Additionally, what happens when a person is over 18 in 1996—is the number negative or are they excluded from the sample?

Another open question with this design, I believe, is whether it is truly receiving the cash payments in childhood, or if it is just receiving the cash payments for a longer period of time, although, again, the degree to which this is correct could be better assessed with some information about the age, time and duration distributions.

The idea that other investments/community resources might explain this association, seems worthy of more exploration/consideration. If the authors can provide any more context on the resources that might have been added over the same period of time that might benefit those who were younger in 1996 versus those who were older, that would be helpful.

In addition, the recession of 2007/8 occurred during the period of outcome measurement. I would suggest the authors discuss how this may or may not relate to their findings.

Finally, because children are all in the same grade when they take the examination, it seems strange that a 3rd order polynomial would be needed for child age. It makes me wonder whether that is picking up other "time" effects that may be currently unaccounted for.

Reviewer #5:

Remarks to the Author:

Report on "Family Cash Transfers and Educational Outcomes in the Next Generations"

The aim of this paper is to test whether cash transfers to families affect children's school performance. They use the establishment of a casino and the following cash transfers to American Indians belonging to the tribe. They specify a difference-in-difference framework pre/post cohorts

affected and using non-American Indians in the same area as a control they use that did not benefit from the transfer. They find a relatively large effect on school outcomes for the children of the affected mothers.

The research question is important, and I applaud the collection of this data. However, I am quite reserved for the "experiment" for cash transfer in this case, and I do not find the empirical specification very convincing. I only discuss some of the main issues, and do not discuss several other issues I have with the paper.

Main issues:

1. One major issue with the interpretation of this as a "natural experiment" is that, more than a cash transfer happens at the same time. For instance, one would expect jobs where made available which means increased earnings etc. So, this is not a clean effect of income transfer to parents, which may affect children. This will of course affect the interpretation as this as an analysis of cash transfer is beneficial to children. For instance, it might be the fact that the American Indians now have a job and income that affect their kids positively, not that they received a transfer.

2. I also have some issue diff in diff set up. The fundamental assumption for using the parallel trend assumption. This means to test for parallel trends between the treated (AI group) and the control (non-AI group) prior to the treatment. Alternatively, one could include linear or non-linear pre-trends. This is not done, and I do not think it can be done with the data available. Since doing this is essential for establishing a causal relationship, it is hard to have trust in the results.

REVIEWER COMMENTS

Reviewer #1 (Remarks to the Author):

This paper reports the intergenerational effects on educational attainment (reading and maths scores) of a cash transfer natural experiment. This is really important work that should inform government programmes. The earlier papers on the first generation effects of this programme (a natural experiment due to casino royalties) have been very influential, and this paper, which goes into the outcomes of the next generation, adds substantively to what was known. I thus recommend publication, and find the analysis, data and reporting to be of good quality.

I have a few comments, which are quite minor.

1. There is no visualization of the results, which is a shame. It seems to me it ought to be quite possible to make some graphs, either from the model predictions or directly from the data, that show how average test score increases in years' exposure for the American Indian but not the other families. This would also allow an intuitive appreciation of the effect size relative to e.g. the AI-nonAI difference, or the secular trend.

We agree with the reviewer that a visualization of the main coefficients could assist with interpretation. We now provide plots of the model-based relation between the exposure and the test score outcome—one for math, and one for reading. We arrived at the plots by using the regression specification which produced Column 2 output of Table 2 (Math) and Table 3 (Reading). Column 2 shows our preferred specification in that it includes as covariates potential confounding variables but excludes potential mediating variables. Next, given that some single years of exposure had few observations (e.g., fewer than 10 AI mothers who were 4 years of age in 1996), we aggregated the continuous exposure variable into the following five categories:

- no exposure before 18 years
- 0 to <3 years of exposure before 18 years of age
- 3 to <6 years of exposure before 18 years of age
- 6 to <9 years of exposure before 18 years of age
- 9+ years of exposure before 18 years of age

We selected these categories to ensure visualization of the exposure / outcome relation across the entire range of exposures while also adhering to the approved human subjects protocol and data use agreement for our project (i.e., not reporting results with cell sizes of <10 observations). Results, shown below, indicate a substantial decline in non-AI test scores among children born to maternal cohorts who were relatively younger in 1996. In addition, the “average” decline by maternal cohort, if one were to aggregate AI and non-AI test scores of their children from this plot, matches the negatively signed *duration* coefficients (i.e., -.041 in Model 2 of Table 2, for Math; and -.043 in Model 2 of Table 3, for Reading). More importantly for our hypothesis, the AI and non-AI test score gap (for both math and reading) narrows with increasing *duration* exposure. Put another way, within the context of the secular decline in 3rd grade test scores in this rural population, the race-based disparity in test scores narrows as mothers of AI children have a relatively greater *duration* exposure.

We now include these two figures in the manuscript.

Figure 3: Third Grade **Math** Z-Score (Fitted Regression Values from Table 2, Model 2) by AI status, by Duration of Exposure to Payments before 18 years.

Figure 4: Third Grade **Reading** Z- Score (Fitted Regression Values from Table 3, Model 2) by AI status, by Duration of Exposure to Payments before 18 years.

2. I found the explanation of the difference in difference regression (lines 146 and also 360) a bit obscure. In fact, it seems to me, this is just a regression model which enters years' exposure, ethnicity and their interaction, with the interaction being the critical test. Or am I missing something? I can't see why this needs to be described as d-in-d at all. I usually associate that term with longitudinal studies where you explain the temporal change in the outcome using temporal change in the predictor. But this study has a one-off outcome and that terms does not apply in quite the same way. In any event, a more intuitive explanation of the modelling strategy is needed (and this would couple well with a figure, see previous point).

The reviewer correctly notes that the interaction between years of exposure and ethnicity serves as the critical test. Given that our research team has backgrounds in public health, psychology, and economics, we debated about which terms to use in describing the study design and analytic strategy. Public health scholars would describe the test as effect modification, psychologists would describe it as moderation, and economists may describe it as a difference-in-difference.

However, based on your and another reviewer's comment, our difference-in-difference text did not clearly describe the approach. We therefore revised the Methods to cohere with the language in psychological sciences, consistent with the notion that the section to which we submitted our manuscript (in *Nature Communications*) is in Human Behaviour. We now refer to the test as "moderation" and focus the description accordingly. Specifically, we examine the influence of the cash transfer by regressing Y (children's test scores) on X1 (time exposed) and X2 (AI status), and then testing whether the relation between Y and X1 is moderated by AI Status (X2). Also, as noted in the previous response, we now couple this description with two additional figures.

3. (Point coming from an Open Science bore). I understand why the authors can only release a de-identified dataset. But rather than saying 'on reasonable request', could this not simply be placed in a repository? 'On request' data sharing is frowned on in my discipline these days because so many authors don't respond to requests, move on, etc.

This comment compelled us to double-check with the North Carolina Education Research Data Center (NCERDC) regarding their policy on data sharing. NCERDC retrieved educational records from the North Carolina Department of Public Instruction (NCDPI) and therefore must adhere to NCDPI regulations as well. Based on our discussions, unfortunately under no circumstances may researchers share data outside of the approved research team. This rule applies to datasets also considered as "de-identified." The rationale relates to assurance of compliance with North Carolina state laws. As it relates to our study of a rural population, the relatively small cell sizes of even aggregate data (e.g., AI mothers in narrow age ranges) could legally be construed as personally identifiable information.

For this reason, we have made two changes. First, we now use the following text regarding data sharing: "Requests for data can be made to the NCERDC." Second, to comply with open science requirements and that of NCERDC, we now post the covariance matrix of the data analyzed along with a vector of means, standard deviations, and number of observations, separately by AI and non-AI participants. This information allows interested

readers to re-create the regression analyses. We now refer readers to this supplemental file. (This supplemental file also provides the summary data points used to create all figures.)

4. In the income and health literature we often find that estimates from interventions are rather different from the estimates you get from observational studies. Though the authors mention that the effects here were about the magnitude you would expect from a government programme in North Carolina, is there more that could be said. For example, you could work out a prior expectation from the observational relationships between childhood income and parental human capital, and then parental human capital and offspring human capital. Or, do we know anything about the educational attainment and income of G2, so we could work out to what extent G3 have 'regressed to the mean', or magnified the different made by the cash transfer in G2. So, all in all, it would be good to situate the effect size relative to what was already known from other designs of study.

We appreciate the Reviewer's encouragement to speculate further on the implications of our results.

Regarding the first suggestion: we first reviewed the literature about the "direct effects" of cash transfers on child educational outcomes. Overall, prior work on cash transfers demonstrates considerable variability in terms of size and strength of expected impacts, with relatively few studies that have focused on learning outcomes specifically (see review by Bastagli and colleagues, 2019). Given that much of this work has been based in low and middle income countries, combined with the small number of estimates available for learning outcomes and test scores, we hesitate to proceed with extrapolations about effect sizes from that broader literature.

The second suggestion—to examine educational attainment of G2 in our study population, and to compare that effect size to that observed for G3— is an intriguing one. Akee and colleagues (2010) studied G2 educational attainment using the Great Smoky Mountains Study cohort. They find that having four more years of "percap" household income increases the child's probability of finishing high school (by 19 yrs) by almost 15 percent. If one instead examined education (in years) continuously, an additional \$4000 per year for the poorest AI households increased educational attainment by one year (by age 21).

The challenge, however, in comparing the magnitude of the high school graduation result (G2) with the test score results (for G3) is two-fold. First, it is not clear in the literature how well graduation maps on to test scores. Second, and more importantly, the casino percap payment included an incentive to finish high school by age 18. If they graduated high school, AI adults became eligible for payment of the semi-annual casino payments themselves; if not, they would have had to wait until age 21. This incentive could have "inflated" graduation rates while not affecting G2 test scores. By contrast, G3 test scores in 3rd grade are not tied to any casino-based financial incentive.

For these reasons, we hesitate to speculate on regression to the mean from G2 effects. That stated, we now note in the revised Discussion that results in G3 appear consistent with a continued benefit across generations of a magnitude that we consider at least moderate.

References cited:

Bastagli, F., Hagen-Zanker, J., Harman, L., Barca, V., Sturge, G. and Schmidt, T., 2019. The impact of cash transfers: a review of the evidence from low-and middle-income countries. *Journal of Social Policy*, 48(3), pp.569-594.

Akee, R.K., Copeland, W.E., Keeler, G., Angold, A. and Costello, E.J., 2010. Parents' incomes and children's outcomes: a quasi-experiment using transfer payments from casino profits. *American Economic Journal: Applied Economics*, 2(1), pp.86-115.

Reviewer #2 (Remarks to the Author):

This article examines effects of casino-related cash gifts on the intergenerational transmission of academic performance in a Native American community. The paper has several strengths, including the multigenerational design, quasi-experimental income manipulation, sample size, cognitive tests, and difference-in-difference statistical models comparing to both non-AI children as well as children born to AI mothers who were relatively older. This is a valuable paper since there is so little multi-generational research on this topic. A few minor comments:

On pg 14 the authors note: "First, the magnitude of the gains to AI children narrows, but does not close, the large AI / non-AI score gap in math (0.46 SD) and reading (0.54 SD)." I think this finding deserves some more thought/speculation, e.g., educational effects of multi-generational discrimination and oppression experienced by AI communities are unlikely to be fully ameliorated by cash gifts.

We thank the reviewer for their comment. We agree that cash transfers alone are unlikely to close the achievement gap considering the historical context of discrimination and oppression faced by AI communities. As a result, we have added additional text to clarify our interpretation of the results. The additional text is as follows:

"First, the magnitude of the gains to AI children depicts a narrowing of the differences between AI and non-AI math and reading scores since the onset of the cash transfer in 1996. Despite the improvements among AI math and reading scores during this 25-year period, the large AI / non-AI score gap in math (0.46 SD) and reading (0.54 SD) scores does not close. The latter is understandable considering cash transfers alone are unlikely to rectify the education effects of well-documented multi-generational discrimination of AI populations (Martinez, 2017). This discrimination includes past and present unequal treatment as well as structural factors that may lead to a higher prevalence of predictors of low educational attainment among AI populations (e.g., poverty, residing near low quality schools, high levels of teen pregnancy; see Demmert and colleagues, 2006)."

References:

Martinez, J. P. (2017). New Mexico's Academic Achievement Gaps: A Synthesis of Status, Causes, and Solutions. A White Paper. *Online Submission*.

Demmert, W. G., Grissmer, D., & Towner, J. (2006). A review and analysis of the research on Native American students. *Journal of American Indian Education*, 5-23.

Throughout, it would help readers if there was more detail on what was considered “younger”, “older”, “longer duration” or note that these are continuous time variables including the range of eg cash transfer duration without having to refer to the Methods section.

We now note in the Results that these are continuous time variables and include, per the Reviewer’s suggestion, the range of cash transfer duration before age 18 years in our data (i.e., from 0 to 15 years’ duration). We also provide a supplemental table which summarizes the range of cash transfer duration, by age (here, we “bin” the table by categories of exposure so as to adhere with IRB regulations of reporting cell sizes for 10 or more observations).

Supplemental Table 1. Counts of AI and non-AI mothers (G2) by years of *duration* exposure, who later gave birth to children who have a recorded 3rd grade test score.

Years of Exposure < 18 years (for G2 mothers)	AI	Non-AI	Age at beginning of exposure (years)
none	252	1825	18+
<3	145	640	15 to 17
3 to <6	176	563	12 to 14
6 to <9	116	349	9 to 11
9 to <12	41	149	6 to 8
12+	10	23	<6

Figure 2 compares reading/math between AI to non-AI groups, but that is not quite the main highlighted finding. Perhaps add a panel that describes the cash duration effect on reading and math.

Done—see two new Figure panels that we added in response to your and another Reviewer’s suggestion.

(repeated response from Reviewer 1’s query) We provide plots of the model-based relation between the exposure and the test score outcome—one for math, and one for reading. We arrived at the plots by using the regression specification which produced Column 2 output of Table 2 (Math) and Table 3 (Reading). Column 2 shows our preferred specification in that it includes as covariates potential confounding variables but excludes potential mediating variables. Next, given that some single years of exposure had few observations (e.g., fewer than 5 AI mothers who were 4 years of age in 1996), we aggregated the continuous exposure variable into the following five categories:

- no exposure before 18 years
- < 3 years of exposure before 18 years of age
- 3 to <6 years of exposure before 18 years of age
- 6 to <9 years of exposure before 18 years of age
- 9+ years of exposure before 18 years of age

We selected these categories to ensure visualization of the exposure / outcome relation across the entire range of exposures while also adhering to the approved human subjects protocol and data use agreement for our project (i.e., not reporting results with cell sizes of <10 observations). Results, shown below, indicate a substantial decline in non-AI test scores among children born to maternal cohorts who were relatively younger in 1996. In addition, the “average” decline by maternal cohort, if one were to aggregate AI and non-AI test scores of their children from this plot, matches the negatively signed *duration* coefficients (i.e., -.041 in Model 2 of Table 2, for Math; and -.043 in Model 2 of Table 3, for Reading). More importantly for our hypothesis, the AI and non-AI test score gap (for both math and reading) narrows with increasing *duration* exposure. Put another way, within the context of the secular decline in 3rd grade test scores in this rural population, the race-based disparity in test scores narrows for AI children whose mothers had a relatively greater *duration* exposure.

We now include these two figures in the manuscript (Figure 3 and 4).

Figure 3: Third Grade **Math** Z-Score (Fitted Regression Values from Table 2, Model 2) by AI status, by Duration of Exposure to Payments before 18 years.

Figure 4: Third Grade **Reading** Z- Score (Fitted Regression Values from Table 3, Model 2) by AI status, by Duration of Exposure to Payments before 18 years.

Reviewer #3

Thank you for the opportunity to review this interesting article examining 2nd generation effects of a natural experiment involving a cash transfer related to the initiation of a casino for American Indian families. The study represents an important addition to the literature regarding the potential benefits of cash transfers, highlighting that potentially particularly for lower resource families the additional cash resources may yield long-term benefits. While there are some limitations to the study, as acknowledged by the authors (in particular the lack of information about fathers' exposure to the cash transfers), on balance the strengths outweigh the weaknesses.

Major:

1. The ages of exposure to cash transfer is hard to clearly understand. The abstract uses the phrase: "during the sensitive period of childhood," though given that childhood may include 0-18 years, that is perhaps an unusually broad set of years to all be classified as "a sensitive period." Given that past work (e.g., Chetty's moving to opportunity analyses) can use quite broad age categories, I wish for greater discussion and the potential for analyses to try to disentangle questions related to the duration of the exposure and the age of exposure (given the possibility that increased plasticity/responsibility may occur at younger ages or alternatively during puberty, specifically).

In light of this comment, we have made two changes. First, we have removed the "sensitive period" language because, as the reviewer notes, we analyze a broad range of child age. Second, our work cannot disentangle "duration" from age in that we do not have AI subjects that received the same duration of cash transfer exposure but who started receiving the transfer at different ages. We, rather, arrive at our exposure variable by using AI status and duration of childhood before 18 years of age exposed to the family cash transfer. We now note this limitation in the Discussion because our work cannot determine which factor (i.e., age at initiation or duration of cash transfer exposure) seems most relevant in designing new interventions.

Minor:

2. Figure 2. It would be helpful to include a color key on the actual figure. Also, consistent font choice and use of black font would enhance readability. I suggest capitalizing "third" for consistency with the other words in the title.
3. Use of a serial comma throughout would enhance clarity
4. I defer to the journal editor regarding whether the written labels in Figure 1 conform to style guides
5. There is inconsistency in whether a space is used (and on both sides) with equal and less than signs

We thank the Reviewer for this attention to detail. We incorporated the suggestions to Figure 2 in that all text is now black and Arial 12 point font. We include a legend in Figure 2 to show the non-AI are the blue bars and AI are the orange bars without the need for readers to refer back to the figure title. We have added a serial comma to the titles of Tables 2 and 3. We have moved the Figure 1 Legend to after the references as indicated

on the Author Guide. We have added a space on both sides of the equal sign throughout the manuscript. We also have added a space on both sides of the less than sign when representing p-values and have omitted a space on the right side of the less than sign for <HS, <37 weeks and <2500 grams. For the latter, the less than sign is used as opposed to writing out less than high school, less than 37 weeks, or less than 2,500 grams. For the former, we are using the less than symbol as conventional when indicating p-values.

Reviewer #4

This paper attempts to identify the impact of receiving a cash infusion for a longer (vs shorter) period of time during childhood on offspring academic test scores, attempting to leverage a large, positive economic shock that was created by the tribal ownership of a casino among the Eastern Band of Cherokee Indians. Payments to tribal members began in 1996 and were issued to all tribal members on a yearly basis thereafter. These payments were about \$5000 a year. The casino and these payments substantially alleviated poverty levels for the tribe. The results could be quite noteworthy as it is difficult to isolate the impacts of additional income, especially income not tied to work. However, the paper currently lacks sufficient detail and robustness checks to bolster confidence in the results and would be substantially improved by providing additional analysis and/or thought around how any unmeasured or collinear biasing factors might impact the results.

We appreciate your constructive comments. In the revised manuscript, we have added more detail about the methods, augmented the results with figures and robustness checks, and added more text regarding the potential role of unmeasured confounding and other potential biases.

While difference-in-difference (DID) designs are very common for evaluating policies or “shocks”, the particular DID approach taken in this paper is a fairly atypical difference-in-difference strategy. Typically, a DID study would compare the change in the outcome for those ‘exposed’ to a shock or policy from before to after “treatment”/shock to the change in an unexposed group over the same period of time. In this paper, the authors are comparing differences in outcomes for four different groups, two of whom have differing durations of exposure to cash payments, and two of whom were never exposed. In essence, we do not observe the “treated” group before “treatment”; we only observe people who have received treatment beginning earlier vs later in childhood.

We thank the Reviewer for her/his/their careful read of the paper. You are correct in that we essentially compare differences in outcomes for four different groups. The Reviewer also notes, along with a separate Reviewer (R1), that a typical DID design would have pre-treatment information on the outcome before the shock. Given that we do not have 3rd grade test scores before the introduction of the casino in this population, and in light of R1’s comments, we modified the description of the methods to no longer refer to the approach as a DID strategy.

The interaction between years of exposure and ethnicity serves as the critical test. Based on your and another reviewer's comment, we therefore revised the Methods to cohere with the language in psychological sciences, consistent with the notion that the section to which we submitted our manuscript (in Nature Communications) is in Human Behaviour. We now refer to the test as "moderation" and focus our description accordingly. Specifically, we examine the influence of the cash transfer by regressing Y (children's test scores) on X1 (time G2 mother is exposed) and X2 (AI status), and then testing whether the relation between Y and X1 is moderated by AI Status (X2).

The identification of the impact of cash for a longer period during childhood relies on the difference in outcomes in 2008-2017 for "younger in 1996" vs "older in 1996" among non-American Indians being a good counterfactual for the difference between "younger in 1996" vs "older in 1996" among American Indians in 2008-2017. Adding some sensitivity or falsification tests seems necessary to bolster the analysis and conclusions. I'm not sure exactly what these could be, partially due to the fact that it has been a very long time since the cash payments started, so it would be difficult to assess pre-policy trends among the different populations. But if the authors could assess an outcome that would conceivably not be responsive to receiving cash for a longer duration of childhood, that would bolster the results. Alternatively, just helping the reader think through what type of bias would need to be operating to invalidate the results would also be helpful.

We agree that our study design uses "older AI in 1996" and non-AI subjects as a series of populations to approximate the hypothetical counterfactual. We also appreciate the reviewer's point that pre-payment 3rd grade test scores (i.e., before 1996) would be useful to assess whether the cash transfers perturbed such scores. We regret that we do not have such test score data, nor do we have educational, health, or human capital outcomes in our dataset that the literature documents as being *unresponsive* to cash transfers (i.e., a falsification check).

That stated, we have augmented our manuscript with several pieces of information which we believe responds to the reviewer and strengthens the argument that results have internal validity. First, we (as described in the subsequent response that directly addresses the age at exposure question) now provide a table of the *duration* exposure variable for AI and non-AI mothers as well as their age distribution in 1996.

Second, in the Supplemental File we now report results in which we restrict the analysis to a narrower range of G2 mothers who received between 0 to 12 years of *duration* exposure by 1996. We arrived at this range of *duration* by inspecting the age distribution of respondents, by AI status and *duration*, and dropping the age groups for which fewer than 10 subjects (AI or non-AI) fell into that cell. We view this sensitivity check as ruling out the possibility that high "outliers" in the *duration* exposure drove results. Findings remain very similar to the original results in column 2 of Tables 2 and 3.

Supplemental Table 2: Restricting the analysis to a narrower range of mothers who received between 0 to 12 years of *duration* exposure by 1996. Regression results predicting Third Grade **Math** Z-score for 4,256 children in Jackson, Swain, and Graham Counties, 2008-2017, as a function of AI race/ethnicity, duration of mother's exposure to family cash transfer as a child, and other covariates.

Variables	Model 2	
	coef	95% CI
AI race/ethnicity (AI) (reference = non-AI)	-0.417****	[-0.518 -0.316]
Duration of potential exposure of mother to family cash transfer in childhood (duration) ¹	-0.042****	[-0.051 -0.033]
Interaction of AI * duration	0.024**	[0.003 0.045]
Offspring Age at test (years)	41.827**	[6.414 77.239]
Offspring Age at test ² (years)	-4.084**	[-7.774 -0.393]
Offspring Age at test ³ (years)	0.130**	[0.002 0.258]
Male sex at birth (reference = female)	0.024	[-0.030 0.079]
	--	--
Intercept included	yes	

¹ This variable indicates number of years of possible receipt of cash transfer for AI participants; for non-AI participants, it represents an age control.

(*p < 0.1, ** p < 0.05, ***p < 0.01, ****p < 0.001)

Supplemental Table 3: Restricting the analysis to a narrower range of mothers who received between 0 to 12 years of *duration* exposure by 1996. Regression results predicting Third Grade **Reading** Z-score for 4,221 children in Jackson, Swain, and Graham Counties, 2008-2017, as a function of AI race/ethnicity, duration of mother's exposure to family cash transfer as a child, and other covariates.

Variables	Model 2	
	coef	95% CI
AI race/ethnicity (AI) (reference = non-AI)	-0.482****	[-0.585 -0.380]
Duration of potential exposure of mother to family cash transfer in childhood (duration) ¹	-0.043****	[-0.052 -0.034]
Interaction of AI * duration	0.024**	[0.003 0.046]
Offspring Age at test (years)	78.193 ****	[41.819 114.566]
Offspring Age at test ² (years)	-7.824****	[-11.619 -4.031]
Offspring Age at test ³ (years)	0.258****	[0.124 0.387]
Male sex at birth (reference = female)	-0.108****	[-0.163 -0.053]
Intercept included	yes	

¹ This variable indicates number of years of possible receipt of cash transfer for AI participants; for non-AI participants, it represents an age control.

(*p < 0.1, ** p < 0.05, ***p < 0.01, ****p < 0.001)

Third, in the Supplemental File we now report results which restrict the mother’s age of delivering children to ages 16 to 35 years. We arrived at this range of maternal age by inspecting the age distribution of mothers at the time of the child’s (G3’s) birth, by AI status, and dropping the maternal ages for which fewer than 10 subjects fell into that cell. We view this sensitivity check as ruling out the possibility that high “outliers” in maternal age, which also (in our case) happen to be classified as “geriatric pregnancies” >35 years and therefore might have independent associations with child test scores, may drive results. Findings remain similar to the original results in columns 2 of Tables 2 and 3, albeit with less precision owing to the dropping of 8% (math) and 11% (reading) of observations after these restrictions.

Supplemental Table 4: Restricting the analysis to a narrower age range of G2 mothers 16 to 35 years at the time of G3’s birth. Regression results predicting Third Grade **Math** Z-score for 3,977 children in Jackson, Swain, and Graham Counties, 2008-2017, as a function of AI race/ethnicity, duration of mother’s exposure to family cash transfer as a child, and other covariates.

Variables	Model 2	
	coef	95% CI
AI race/ethnicity (AI) (reference = non-AI)	-0.390***	[-0.494 -0.248]
Duration of potential exposure of mother to family cash transfer in childhood (duration) ¹	-0.038****	[-0.048 -0.029]
Interaction of AI * duration	0.021*	[-0.0005 0.042]
Offspring Age at test (years)	47.954**	[11.73 84.17]
Offspring Age at test ² (years)	-4.728**	[-8.50 -0.955]
Offspring Age at test ³ (years)	0.153**	[0.022 0.284]
Male sex at birth (reference = female)	0.017	[-0.040 0.074]
Intercept included	yes	

¹ This variable indicates number of years of possible receipt of cash transfer for AI participants; for non-AI participants, it represents an age control.

(*p < 0.1, ** p < 0.05, ***p < 0.01, ****p < 0.001)

Supplemental Table 5: Restricting the analysis to a narrower age range of G2 mothers 16 to 35 years at the time of G3's birth. Regression results predicting Third Grade Reading Z-score for 3,944 children in Jackson, Swain, and Graham Counties, 2008-2017, as a function of AI race/ethnicity, duration of mother's exposure to family cash transfer as a child, and other covariates.

Variables	Model 2	
	coef	95% CI
AI race/ethnicity (AI) (reference = non-AI)	-0.452****	[-0.558 -0.346]
Duration of potential exposure of mother to family cash transfer in childhood (duration) ¹	-0.040****	[-0.049 -0.031]
Interaction of AI * duration	0.022**	[0.0002 0.043]
Offspring Age at test (years)	86.932****	[49.753 124.11]
Offspring Age at test ² (years)	-8.738****	[-12.613 -4.863]
Offspring Age at test ³ (years)	0.290****	[0.124 0.387]
Male sex at birth (reference = female)	-0.094***	[-0.151 -0.0365]
Intercept included	yes	

¹ This variable indicates number of years of possible receipt of cash transfer for AI participants; for non-AI participants, it represents an age control.

(*p < 0.1, ** p < 0.05, ***p < 0.01, ****p < 0.001)

Fourth, and perhaps most importantly, we now (in the revised Discussion) walk the reader through sources of bias that would need to operate in order to invalidate results. We also give two examples of potential confounders and describe why we do not view this confounding as a plausible rival explanation for the pattern of results.

“Within the context of the secular decline in 3rd grade test scores in this rural population, the AI / non-AI disparity in test scores narrows as mothers of AI children have a relatively greater *duration* exposure. Whereas we infer that this finding arises from the benefit of the cash transfer to AI families, we cannot rule out the possibility of unmeasured confounding. Such a confounder would have to correlate positively with our exposure (but not be caused by it), occur only among AI families (but not among non-AI families), and vary positively with 3rd grade test scores. School-based investments particular to AI children that concentrate in recent years, or broader employment gains to AI families that concentrate in recent years, could meet these criteria. We, however, know of no such trend in school-based investments unique to AI children in public schools (i.e., the datasets we were able to link for our tests). In addition, both AI and non-AI adults show employment gains following the opening of the casino, which minimizes the plausibility that this factor introduces bias.”

As above, the main strategy for assessing impacts of receiving cash payments for a longer (vs shorter) duration is to compare academic outcomes during 2008-2017 for American Indian children of birthing people who would have received cash payments for a longer duration by

virtue of being relatively younger in 1996 when the payments began to outcomes for American Indian Children born to birthing people who were older in 1996 and then additionally compare the difference in outcomes for these two populations to the difference in outcomes for non-American Indian birthing people who were younger in 1996 to non-American Indian people who were older in 1996. Because the identification is linked to being older or younger in one calendar year, the issue of time in all its forms (age, year, cohort) seem particularly important to pay attention to, even if just by explaining to the reader what assumptions are being relied on. Currently, the paper does not do enough to subject the results to robustness checks and falsification tests that would help the reader feel more confident that the results are indeed due to a longer duration of casino payments during the childhood. Below I suggest some additional information that would help probe the results and assess the degree to which they may or may not be due to other factors.

More information is required to explain the age distribution of the sample and overlap in maternal age for people exposed at younger versus older ages to the beginning of the cash transfers. A table of how many AI and non-AI are in different ‘bins’ of duration for each year of the outcome data would be very helpful for visualizing overlap. I think the analysis should be limited to women with maternal ages for which there is good overlap for those who received the payments for a longer duration during childhood versus a shorter duration.

Per your suggestion, this table (Supplemental Table 1, which we place in the Supplement) provides the counts of AI and non-AI mothers (G2) by years of *duration* exposure, who later gave birth to children who have a recorded 3rd grade test score:

Supplemental Table 1. Counts of AI and non-AI mothers (G2) by years of *duration* exposure, who later gave birth to children who have a recorded 3rd grade test score

Years of Exposure < 18 years (for G2 mothers)	AI	Non-AI	Age at beginning of exposure (years)
none	252	1825	18+
<3	145	640	15 to 17
3 to <6	176	563	12 to 14
6 to <9	116	349	9 to 11
9 to <12	41	149	6 to 8
12+	10	23	<6

We binned the exposure similarly to the binning categories used for the additional Figure panels we now provide in the revision (per the suggestion of another reviewer). Based on this table, we do not run into “small cell” sizes until the 12+ years exposure category. This circumstance makes sense in that, as the reviewer points out, the cohort of mothers who are 6 years of age or less in 1996 will not have many reproductive years to have a child who reaches 3rd grade by 2017 (i.e., the last year for which we could link test score data to birth records). Put another way, much of the reproductive-age lifespan of these future mothers will not have yet appeared in terms of births recorded in vital statistics data. We now acknowledge this age-cohort constraint in the revised Discussion.

The Reviewer also recommends that we restrict the analysis to maternal ages in which there is sufficient AI / non-AI overlap. We have, as described in the previous response, performed two supplemental analyses in which we (1) remove (G2) observations from the 12+ exposure category; and (2) restrict to maternal ages of 16 to 35 years at the time of the child's (G3's) birth (these age ranges have good overlap for AI / non-AI). In these analyses, inference for both reading and math scores remains similar to the original test.

From my calculations, it seems like there would actually be very few people in the sample who would be under 5 when cash payments began and have children in third grade between 2008 to 2017. Specifically, birthing people who are 5 in 1996 will be 26 in 2017. If third graders are approximately 8 years old, they would have had children at 18, which is not highly unusual, but would be a relatively small portion of the sample. Seems that even fewer would have children before that. This has at least 2 implications: This means that effects we are seeing are due to receiving cash for a longer period in childhood, but still largely outside of the birth to five range in which previous literature has shown that investments produce large returns to health and educational outcomes.

This is an important point which we failed to mention in the original manuscript. We now note this implication in a new paragraph in the revised Discussion:

“Many AI (G2) mothers who were very young in 1996 (i.e., < 5 years old) have children that are scheduled to attend third grade after 2017—the last year in which we could link test score information. This circumstance means that our analysis includes very few (G2) mothers who had early-life exposure (i.e., from infancy to age 5) to the cash transfer. Our results may therefore underestimate the potentially larger benefit of cash transfers (especially before age 5 years among G2) that may accrue to the subsequent generation of AI children and produce large returns to health and education (Carneiro and Heckman, 2003; Duncan and colleagues, 2014).”

References:

Carneiro, P. M., & Heckman, J. J. (2003). Human capital policy.

Duncan, G. J., Magnuson, K., & Votruba-Drzal, E. (2014). Boosting family income to promote child development. *The Future of Children*, 99-120.

Second, people who were younger vs older in 1996 are also going to be different on either maternal age at birth or the year their children are in third grade. I do not think there is a way to disentangle these potentially different effects, both of which could impact test scores. Someone who was 5 in 1996 and had a child at 18 would have third grade test scores for their children in 2017, whereas someone who was 10 in 1996 and had a child at 18 would have third grade test scores in 2012. There could be secular trends in test scores over this period and the assumption of the models are that these are the same for AI versus non-AI children. While some aspects may not be directly testable, if the authors discuss these limitations and assumptions and are able to reason that, if anything, we might expect effects in the opposite direction than what we are seeing, that could also bolster the arguments.

The Reviewer raises two related points, which we address in turn.

The first point relates to the fact that younger vs. older mothers in 1996 will be different on maternal age at birth. We agree, and if there is not adequate age overlap between AI and non-AI mothers, then non-AI mothers may not appropriately serve as a comparison group. For this reason, we now report a supplemental analysis in which we restrict to mothers of overlapping ages (i.e., 16 to 35 years) and find similar results to the original test. In addition, although delayed maternal age could serve as a mechanism through which the cash transfer affects future children's test scores, the model 3 columns of Tables 2 and 3 do not indicate that maternal age shows an association with children's test scores. This circumstance renders the explanation of non-overlapping maternal ages as less plausible in driving the results.

The second point relates to the notion that secular trend in test scores could drive the association. The plots below show mean math and reading test scores (normalized annually by Z-score), by year and AI/non-AI status. None shows any notable trend. We also, in a separate response to the reviewer, show regression results which control for test year (both as a continuous variable and with year fixed-effects). Inference remains very similar to the original tests, which indicates that any temporal pattern in test scores does not drive results.

Additionally, I think that people who were younger (vs older) in 1996 would have received the cash payments for a longer total period of time, not just in childhood (although I'm not completely sure about this; this might be taken care of through the wide interval of outcome data collection 2008-2017).

The reviewer is correct because cash payments occur every year after 1996. We arrive at our exposure variable by using AI status and duration of childhood before 18 years of age exposed to the family cash transfer. We now note a key limitation in the Discussion in that our work cannot determine which factor (i.e., age at initiation or duration of cash transfer exposure) seems most relevant. As the reviewer points out, however, mothers of a wide range of ages give birth to children who are captured in the wide interval of 3rd grade test score data (collected from 2008 to 2017). This circumstance gives us ample variation in our exposure of interest (also shown in our Supplemental Table 1 [above]).

In a related comment, the time period for which outcomes are compared is 2008 to 2017 is a pretty long period of time. Are there trends in test scores over this period, are the trends parallel for AI and non-AI. It seems like the identification strategy needs to have all 4 groups – younger in 1996 AI, older in 1996 AI, younger in 1996 non-AI, older in 1996 non-AI in each or most of these outcome years. If you do, than it would seem you could include year fixed effects or at least year trends. However, including year fixed effects would, I think by structure, mean that you are comparing people of different maternal ages, so limiting to a subset of maternal ages might be necessary, at least as a sensitivity analysis.

Per an earlier response, we provide a plot of the test scores, by race/ethnicity, over time. This plot indicates no discernable trend. We also provide summary regression results below, for math and reading, in which we performed the additional controls for time (continuous time, and separately, year fixed effects with a restricted maternal age range) suggested by the reviewer. For brevity, we show only the regression coefficients for AI, *duration*, AI**duration* and test year, but all other covariates from Model 2 of original Tables 2 and 3 are included in the model. Inference for the AI**duration* coefficient remains the same as in the original test—and in some specifications, the result becomes slightly stronger. Taken together, these robustness checks indicate that secular patterns in test scores cannot account for our observed results.

Supplemental Table 6: Control for test year (continuous). Regression results predicting Third Grade **Math** Z-score for 4,289 children in Jackson, Swain, and Graham Counties, 2008-2017, as a function of AI race/ethnicity, duration of mother’s exposure to family cash transfer as a child, and other covariates.

Variables [‡]	Model 2	
	coef	95% CI
AI race/ethnicity (AI) (reference = non-AI)	-0.423***	[-0.523 -0.323]
Duration of potential exposure of mother to family cash transfer in childhood (duration) ¹	-0.038****	[-0.048 -0.028]
Interaction of AI * duration	0.025**	[0.005 0.045]
Test year (continuous)	-0.008	[-0.018 0.003]

¹ This variable indicates number of years of possible receipt of cash transfer for AI participants; for non-AI participants, it represents an age control.

[‡] child age and gender controls included in regression but not shown here (full results for covariates available in data repository).

(*p < 0.1, ** p < 0.05, ***p < 0.01, ****p < 0.001)

Supplemental Table 7: Control for test year (continuous). Regression results predicting Third Grade **Reading** Z-score for 4,254 children in Jackson, Swain, and Graham Counties, 2008-2017, as a function of AI race/ethnicity, duration of mother's exposure to family cash transfer as a child, and other covariates.

Variables [‡]	Model 2	
	coef	95% CI
AI race/ethnicity (AI) (reference = non-AI)	-0.493***	[-0.594 -0.392]
Duration of potential exposure of mother to family cash transfer in childhood (duration) ¹	-0.043****	[-0.053 -0.033]
Interaction of AI * duration	0.028***	[0.008 0.049]
Test year (continuous)	0.0006	[-0.010 0.012]

¹ This variable indicates number of years of possible receipt of cash transfer for AI participants; for non-AI participants, it represents an age control.

[‡] child age and gender controls included in regression but not shown here (full results for covariates available in data repository).

(*p < 0.1, ** p < 0.05, ***p < 0.01, ****p < 0.001)

Supplemental Table 8: Including Test Year Fixed Effects and restricting the analysis to a narrower age range of G2 mothers 16 to 35 years at the time of G3's birth. Regression results predicting Third Grade **Math** Z-score for 3,977 children in Jackson, Swain, and Graham Counties, 2008-2017, as a function of AI race/ethnicity, duration of mother's exposure to family cash transfer as a child, and other covariates.

Variables [‡]	Model 2	
	coef	95% CI
AI race/ethnicity (AI) (reference = non-AI)	-0.376***	[-0.480 -0.272]
Duration of potential exposure of mother to family cash transfer in childhood (duration) ¹	-0.038****	[-0.049 -0.028]
Interaction of AI * duration	0.027**	[0.006 0.048]
Test year fixed effects	Included; not shown	

¹ This variable indicates number of years of possible receipt of cash transfer for AI participants; for non-AI participants, it represents an age control.

[‡] child age and gender controls included in regression but not shown here (full results for covariates available in data repository).

(*p < 0.1, ** p < 0.05, ***p < 0.01, ****p < 0.001)

Supplemental Table 9: Including Test Year Fixed Effects and restricting the analysis to a narrower age range of G2 mothers 16 to 35 years at the time of G3's birth. Regression results predicting Third Grade **Reading** Z-score for 3,944 children in Jackson, Swain, and Graham Counties, 2008-2017, as a function of AI race/ethnicity, duration of mother's exposure to family cash transfer as a child, and other covariates.

Variables [‡]	Model 2	
	coef	95% CI
AI race/ethnicity (AI) (reference = non-AI)	-0.443***	[-0.549 -0.337]
Duration of potential exposure of mother to family cash transfer in childhood (duration) ¹	-0.040****	[-0.051 -0.030]
Interaction of AI * duration	0.024**	[0.002 0.046]
Test year fixed effects	Included; not shown	

¹ This variable indicates number of years of possible receipt of cash transfer for AI participants; for non-AI participants, it represents an age control.

[‡] child age and gender controls included in regression but not shown here (full results for covariates available in data repository).

(*p < 0.1, ** p < 0.05, ***p < 0.01, ****p < 0.001)

More details of the calculation of duration are needed. Specifically, although it is implied through the naming of the variable as 'duration', it should be clearly stated whether each participant's age in 1996 is subtracted from 18 or the other way around. Additionally, what happens when a person is over 18 in 1996—is the number negative or are they excluded from the sample?

We believe that our new text, tables, and figures (described above) now clarify the specifics about the calculation of *duration*. We also now add text noting that a person over 18 in 1996 receives a "0" *duration* value and that we retain them in the sample.

Another open question with this design, I believe, is whether it is truly receiving the cash payments in childhood, or if it is just receiving the cash payments for a longer period of time, although, again, the degree to which this is correct could be better assessed with some information about the age, time and duration distributions.

We now note this limitation in the Discussion in that our work cannot disentangle which factor (i.e., age at initiation or duration of cash transfer exposure) affects AI test scores.

The idea that other investments/community resources might explain this association, seems worthy of more exploration/consideration. If the authors can provide any more context on the resources that might have been added over the same period of time that might benefit those who were younger in 1996 versus those who were older, that would be helpful.

There were other effects of the casino opening besides the cash transfer to tribal members. The tribe designated half of the gaming revenues for individual tribal members and half to community investments, including behavioral health, drug abuse prevention, health care, education, and social services.¹ In addition, the casino itself is the largest employer in the region and boosts other local businesses (see article on 2022 Economic Impact report: <https://casinosnc.com/harrahs-chokeee-releases-2022-economic-impact-report>). For example, in the 2011 University of North Carolina assessment of the casino's impact, the casino appears to have attenuated seasonal fluctuations in unemployment, raised the employment rate in Jackson and Swain counties, and increased wages and salaries.¹ A number of these economic and noneconomic factors together may have contributed to an increase in the social cohesion of the community itself.² These impacts might be associated with potentially improved health and functioning for all American Indians as well as other individuals in the study region.

Each of these other economic (and noneconomic) factors, described above, are not age dependent and would therefore affect AI participants similarly. In addition, we know of no public-school based improvements over time that would affect AI children but not non-AI children.

In the revision, we now insert a paragraph in the Discussion which provides this community context of the cash transfer and the opening of the casino.

References

1. Johnson JH, Kasarda JD, Appold SJ. *Assessing the Economic and Non-Economic Impacts of Harrah's Cherokee Casino*. Enterprise. FHKIoP; 2017.
2. Bullock A, Bradley VL. Family income supplements and development of psychiatric and substance use disorders among an American Indian population. *JAMA*. 2010;304(9):962-963.

In addition, the recession of 2007/8 occurred during the period of outcome measurement. I would suggest the authors discuss how this may or may not relate to their findings.

Given that the Great Recession began during the first year of outcome measurement, one challenge is that we could not examine the influence of this ecological "shock." However, to the extent that our year fixed-effects specification (supplemental analysis) controls for the unique year effects (and results remain essentially similar to the original tests), we view it as unlikely that the Great Recession biases our results. We now insert short text that refers to the Great Recession.

As an aside: this rural population was likely exposed to a number of subsequent economic shocks, making it (in our view) more salient that we observe persistent associations of the cash transfer across both time and generations.

Finally, because children are all in the same grade when they take the examination, it seems strange that a 3rd order polynomial would be needed for child age. It makes me wonder whether that is picking up other "time" effects that may be currently unaccounted for.

We suspect that the polynomial best fits the data for two reasons. First, within the normative 1-year age range of 3rd graders, being slightly older confers a benefit on test score performance. Much literature supports this claim. Second, persons who are substantially older than the “normative” age range could represent children who are held back by either the school or the parent for reasons that correlate with lower test performance.

We also note that standardizing test scores, for each year, to a Z-score minimizes the across year variations in test scores that often arise due to many factors (e.g., the changing questions that appear on the state tests, the difference in level of readiness across cohorts).

Reviewer #5

The aim of this paper is to test whether cash transfers to families affect children’s school performance. They use the establishment of a casino and the following cash transfers to American Indians belonging to the tribe. They specify a difference-in-difference framework pre/post cohorts affected and using non-American Indians in the same area as a control they use that did not benefit from the transfer. They find a relatively large effect on school outcomes for the children of the affected mothers.

The research question is important, and I applaud the collection of this data. However, I am quite reserved for the “experiment” for cash transfer in this case, and I do not find the empirical specification very convincing. I only discuss some of the main issues, and do not discuss several other issues I have with the paper.

Main issues:

1. One major issue with the interpretation of this as a “natural experiment” is that, more than a cash transfer happens at the same time. For instance, one would expect jobs were made available which means increased earnings etc. So, this is not a clean effect of income transfer to parents, which may affect children. This will of course affect the interpretation as this as an analysis of cash transfer is beneficial to children. For instance, it might be the fact that the American Indians now have a job and income that affect their kids positively, not that they received a transfer.

As the reviewer notes, there were other effects of the casino opening besides the cash transfer to tribal members. Indeed, the tribe designated half of the gaming revenues for individual tribal members and half to community investments, including behavioral health, drug abuse prevention, health care, education, and social services.¹ In addition, the casino itself is the largest employer in the region and boosts other local businesses (see article on 2022 Economic Impact report: <https://casinosnc.com/harrahs-chokechee-releases-2022-economic-impact-report>). For example, in the 2011 University of North Carolina assessment of the casino’s impact, the casino appears to have attenuated seasonal fluctuations in unemployment, raised the employment rate in Jackson and Swain counties, and increased wages and salaries.¹ A number of these economic and noneconomic factors together may have contributed to an increase in the social cohesion of the

community itself.² These impacts might be associated with potentially improved health and functioning for all American Indians as well as other individuals in the study region.

It is for this reason that we tested an *interaction term* between maternal age in 1996 and American Indian status. Our model is specifically testing whether there are differences on the basis of maternal age in 1996 rather than American Indian status as a main effect. Each of these other economic (and noneconomic) factors are not age dependent and would therefore affect American Indian participants similarly. Our pattern of results suggests that the beneficial association is related to the maternal age at the start of the cash transfer for American Indian participants (and therefore the duration of the exposure to the family cash transfer).

We now insert, in the Discussion, this paragraph which describes the community context and the general accrual of community benefits to the rural population.

1. Johnson JH, Kasarda JD, Appold SJ. *Assessing the Economic and Non-Economic Impacts of Harrah's Cherokee Casino*. Enterprise. FHKloP; 2017.
2. Bullock A, Bradley VL. Family income supplements and development of psychiatric and substance use disorders among an American Indian population. *JAMA*. 2010;304(9):962-963.

2. I also have some issue diff in diff set up. The fundamental assumption for using the parallel trend assumption. This means to test for parallel trends between the treated (AI group) and the control (non-AI group) prior to the treatment. Alternatively, one could include linear or non-linear pre-trends. This is not done, and I do not think it can be done with the data available. Since doing this is essential for establishing a causal relationship, it is hard to have trust in the results.

Thank you for this comment. We have made four key changes which we believe substantially improves the likelihood that readers will “trust” the internal validity of the results.

First, we no longer refer to the analysis as a DID approach, given the fact that we do not have pre-1996 data on 3rd grade test scores. We therefore revised the Methods to cohere with the language in psychological sciences, consistent with the notion that the section to which we submitted our manuscript (in Nature Communications) is in Human Behaviour. We now refer to the test as “moderation” and focus the description accordingly. Specifically, we examine the influence of the cash transfer by regressing Y (children’s test scores) on X1 (time exposed) and X2 (AI status), and then testing whether the relation between Y and X1 is moderated by AI Status (X2).

Second, we conduct a series of robustness checks to determine whether results appear sensitive to outliers in either the *duration* exposure variable or to outliers in maternal age at the time of G3 infant’s birth. Inference for our tests (for both math and reading scores) remains the same as in the original Tables (see below).

Third, we included year fixed effects (i.e., indicator variables for year 3rd grade tests were taken) as a robustness check to rule out the possibility that temporal trends in test scores, which may correlate with but not be caused by the *duration* exposure, drive results. Findings remain essentially unchanged (see below).

Fourth, and perhaps most importantly, we now (in the revised Discussion) walk the reader through sources of bias that would need to operate in order to invalidate results. We also give two examples of potential confounders and describe why we do not view this confounding as a plausible rival explanation for the pattern of results:

“Within the context of the secular decline in third grade test scores in this rural population, the AI / non-AI disparity in test scores narrows as mothers of AI children have a relatively greater *duration* exposure. Whereas we infer that this finding arises from the benefit of the cash transfer to AI families, we cannot rule out the possibility of unmeasured confounding. Such a confounder would have to correlate positively with our exposure (but not be caused by it), occur only among AI families (but not among non-AI families), and vary positively with third grade test scores. School-based investments particular to AI children that concentrate in recent years, or broader employment gains to AI families that concentrate in recent years, could meet these criteria. We, however, know of no such trend in school-based investments unique to AI children in public schools. In addition, both AI and non-AI adults show employment gains following the opening of the casino, which minimizes the plausibility that this factor introduces bias.”

If the Reviewer, moreover, has other specific suggestions to strengthen assessments of internal validity, we welcome these suggestions.

Supplemental Table 2: Restricting the analysis to a narrower range of mothers who received between 0 to 12 years of *duration* exposure by 1996. Regression results predicting Third Grade Math Z-score for 4,256 children in Jackson, Swain, and Graham Counties, 2008-2017, as a function of AI race/ethnicity, duration of mother’s exposure to family cash transfer as a child, and other covariates.

Variables	Model 2	
	coef	95% CI
AI race/ethnicity (AI) (reference = non-AI)	-0.417****	[-0.518 -0.316]
Duration of potential exposure of mother to family cash transfer in childhood (duration) ¹	-0.042****	[-0.051 -0.033]
Interaction of AI * duration	0.024**	[0.003 0.045]
Offspring Age at test (years)	41.827**	[6.414 77.239]
Offspring Age at test ² (years)	-4.084**	[-7.774 -0.393]
Offspring Age at test ³ (years)	0.130**	[0.002 0.258]
Male sex at birth (reference = female)	0.024	[-0.030 0.079]
	--	--
Intercept included	yes	

¹ This variable indicates number of years of possible receipt of cash transfer for AI participants; for non-AI participants, it represents an age control.
 (*p < 0.1, ** p < 0.05, ***p < 0.01, ****p < 0.001)

Supplemental Table 3: Restricting the analysis to a narrower range of mothers who received between 0 to 12 years of *duration* exposure by 1996. Regression results predicting Third Grade **Reading** Z-score for 4,221 children in Jackson, Swain, and Graham Counties, 2008-2017, as a function of AI race/ethnicity, duration of mother's exposure to family cash transfer as a child, and other covariates.

Variables	Model 2	
	coef	95% CI
AI race/ethnicity (AI) (reference = non-AI)	-0.482****	[-0.585 -0.380]
Duration of potential exposure of mother to family cash transfer in childhood (duration) ¹	-0.043****	[-0.052 -0.034]
Interaction of AI * duration	0.024**	[0.003 0.046]
Offspring Age at test (years)	78.193 ****	[41.819 114.566]
Offspring Age at test ² (years)	-7.824****	[-11.619 -4.031]
Offspring Age at test ³ (years)	0.258****	[0.124 0.387]
Male sex at birth (reference = female)	-0.108****	[-0.163 -0.053]
Intercept included	yes	

¹ This variable indicates number of years of possible receipt of cash transfer for AI participants; for non-AI participants, it represents an age control.

(*p < 0.1, ** p < 0.05, ***p < 0.01, ****p < 0.001)

Supplemental Table 4: Restricting the analysis to a narrower age range of G2 mothers 16 to 35 years at the time of G3's birth. Regression results predicting Third Grade **Math** Z-score for 3,977 children in Jackson, Swain, and Graham Counties, 2008-2017, as a function of AI race/ethnicity, duration of mother's exposure to family cash transfer as a child, and other covariates.

Variables	Model 2	
	coef	95% CI
AI race/ethnicity (AI) (reference = non-AI)	-0.390***	[-0.494 -0.248]
Duration of potential exposure of mother to family cash transfer in childhood (duration) ¹	-0.038****	[-0.048 -0.029]
Interaction of AI * duration	0.021*	[-0.0005 0.042]
Offspring Age at test (years)	47.954**	[11.73 84.17]
Offspring Age at test ² (years)	-4.728**	[-8.50 -0.955]
Offspring Age at test ³ (years)	0.153**	[0.022 0.284]
Male sex at birth (reference = female)	0.017	[-0.040 0.074]
Intercept included	yes	

¹ This variable indicates number of years of possible receipt of cash transfer for AI participants; for non-AI participants, it represents an age control.

(*p < 0.1, ** p < 0.05, ***p < 0.01, ****p < 0.001)

Supplemental Table 5: Restricting the analysis to a narrower age range of G2 mothers 16 to 35 years at the time of G3's birth. Regression results predicting Third Grade Reading Z-score for 3,944 children in Jackson, Swain, and Graham Counties, 2008-2017, as a function of AI race/ethnicity, duration of mother's exposure to family cash transfer as a child, and other covariates.

Variables	Model 2	
	coef	95% CI
AI race/ethnicity (AI) (reference = non-AI)	-0.452****	[-0.558 -0.346]
Duration of potential exposure of mother to family cash transfer in childhood (duration) ¹	-0.040****	[-0.049 -0.031]
Interaction of AI * duration	0.022**	[0.0002 0.043]
Offspring Age at test (years)	86.932****	[49.753 124.11]
Offspring Age at test ² (years)	-8.738****	[-12.613 -4.863]
Offspring Age at test ³ (years)	0.290****	[0.124 0.387]
Male sex at birth (reference = female)	-0.094***	[-0.151 -0.0365]
Intercept included	yes	

¹ This variable indicates number of years of possible receipt of cash transfer for AI participants; for non-AI participants, it represents an age control.

(*p < 0.1, ** p < 0.05, ***p < 0.01, ****p < 0.001)

Supplemental Table 8: Including Test Year Fixed Effects and restricting the analysis to a narrower age range of G2 mothers 16 to 35 years at the time of G3's birth. Regression results predicting Third Grade **Math** Z-score for 3,977 children in Jackson, Swain, and Graham Counties, 2008-2017, as a function of AI race/ethnicity, duration of mother's exposure to family cash transfer as a child, and other covariates.⁺

Variables	Model 2	
	coef	95% CI
AI race/ethnicity (AI) (reference = non-AI)	-0.376***	[-0.480 -0.272]
Duration of potential exposure of mother to family cash transfer in childhood (duration) ¹	-0.038****	[-0.049 -0.028]
Interaction of AI * duration	0.027**	[0.006 0.048]
Test year fixed effects	Included; not shown	

¹ This variable indicates number of years of possible receipt of cash transfer for AI participants; for non-AI participants, it represents an age control.

⁺ other covariates include child sex and age, age-squared, and age-cubed (output not shown).

(*p < 0.1, ** p < 0.05, ***p < 0.01, ****p < 0.001)

Supplemental Table 9: Including Test Year Fixed Effects and restricting the analysis to a narrower age range of G2 mothers 16 to 35 years at the time of G3's birth. Regression results predicting Third Grade **Reading** Z-score for 3,944 children in Jackson, Swain, and Graham Counties, 2008-2017, as a function of AI race/ethnicity, duration of mother's exposure to family cash transfer as a child, and other covariates.⁺

Variables	Model 2	
	coef	95% CI
AI race/ethnicity (AI) (reference = non-AI)	-0.443***	[-0.549 -0.337]
Duration of potential exposure of mother to family cash transfer in childhood (duration) ¹	-0.040****	[-0.051 -0.030]
Interaction of AI * duration	0.024**	[0.002 0.046]
Test year fixed effects	Included; not shown	

¹ This variable indicates number of years of possible receipt of cash transfer for AI participants; for non-AI participants, it represents an age control.

‡ child age and gender controls included in regression but not shown here (full results for covariates available in data repository).

⁺ other covariates include child sex and age, age-squared, and age-cubed (output not shown).

(*p < 0.1, ** p < 0.05, ***p < 0.01, ****p < 0.001)

Reviewers' Comments:

Reviewer #2:

Remarks to the Author:

All my concerns have been addressed and the edits in response to my and the other reviews' suggestions improve the quality of the paper. Thus I suggest publication.

Reviewer #3:

Remarks to the Author:

The authors have responded to the initial comments by including two new figures that illustrate estimated reading scores as a function of years of exposure to cash transfers among American Indian (AI) and non-AI individuals. While I appreciate the effort to enhance the manuscript with these visual aids, I find myself deeply confused about the results and their interpretation.

The core of my confusion lies in the interpretation of the impact of cash transfers on reading scores. The authors suggest that cash transfers have "improved math and reading third grade test scores." However, the coefficients for change in both math reading scores indicate a decrease (worsening) in scores with more years of cash transfers in the prior generation, which is also apparent in the newly shared figures (3-4). This trend is particularly perplexing when considering that the gap between AI and non-AI individuals is closing not because of an improvement in the AI group, but rather due to a significant decline in the non-AI group, who, notably, did not directly receive these cash transfers.

This leads to a critical question: Is the story here that cash transfers do not benefit the offspring of the recipients in certain domains, and might actually have adverse effects on the offspring of non-recipients in the community? This hypothesis is intriguing, yet I struggle to conceive a mechanism for this potential harm, especially given that the non-AI group did not receive the intervention.

Another possibility that merits consideration is the validity of the measures of math and reading ability used in the study. Given the potential for variability in state tests (due to changes in the test itself, the thresholds, or teaching methods), it is plausible that the observed patterns may not accurately reflect true changes in academic abilities. This aspect deserves further scrutiny and discussion, as it could significantly impact the interpretation of the results.

In conclusion, while the addition of the figures provides a visual representation of the data, it raises important questions about the interpretation of the results and the validity of the measures used. I suggest that the authors provide a more detailed examination of these aspects to clarify the implications of their findings.

Reviewer #4:

Remarks to the Author:

The authors have adequately addressed all of my previous questions and concerns.

REVIEWER COMMENTS

Reviewer #2:

All my concerns have been addressed and the edits in response to my and the other reviews' suggestions improve the quality of the paper. Thus I suggest publication.

Thank you.

Reviewer #3:

The authors have responded to the initial comments by including two new figures that illustrate estimated reading scores as a function of years of exposure to cash transfers among American Indian (AI) and non-AI individuals. While I appreciate the effort to enhance the manuscript with these visual aids, I find myself deeply confused about the results and their interpretation.

The core of my confusion lies in the interpretation of the impact of cash transfers on reading scores. The authors suggest that cash transfers have "improved math and reading third grade test scores." However, the coefficients for change in both math reading scores indicate a decrease (worsening) in scores with more years of cash transfers in the prior generation, which is also apparent in the newly shared figures (3-4). This trend is particularly perplexing when considering that the gap between AI and non-AI individuals is closing not because of an improvement in the AI group, but rather due to a significant decline in the non-AI group, who, notably, did not directly receive these cash transfers.

This leads to a critical question: Is the story here that cash transfers do not benefit the offspring of the recipients in certain domains, and might actually have adverse effects on the offspring of non-recipients in the community? This hypothesis is intriguing, yet I struggle to conceive a mechanism for this potential harm, especially given that the non-AI group did not receive the intervention.

Another possibility that merits consideration is the validity of the measures of math and reading ability used in the study. Given the potential for variability in state tests (due to changes in the test itself, the thresholds, or teaching methods), it is plausible that the observed patterns may not accurately reflect true changes in academic abilities. This aspect deserves further scrutiny and discussion, as it could significantly impact the interpretation of the results.

In conclusion, while the addition of the figures provides a visual representation of the data, it raises important questions about the interpretation of the results and the validity of the measures used. I suggest that the authors provide a more detailed examination of these aspects to clarify the implications of their findings.

The reviewer raises several important points regarding the interpretation of results and the validity of the data used. We address each of these points below.

In light of the Reviewer's suggestion that we better align the text description of the Results with the new Figures, we now make clear in the Results and Discussion that, within the context of relatively lower third grade test scores in this rural population, the race-based disparity in test scores narrows for AI children whose mothers had a relatively greater *duration* exposure. This conclusion differs from any declaration that AI children with a greater duration of exposure show higher absolute levels of math and reading test scores than do any other group. We also now removed in the revision any text that could be misconstrued to imply such a conclusion.

Next, the Reviewer points out that the narrowing of the test score gap between AI and non-AI children, as a function of increased duration of exposure, appears to occur owing to a more rapid reduction of test scores among non-AI (vs. AI) children (Figures 3 and 4). We, like the Reviewer, know of no reason to suspect that the accumulated cash transfers to AI persons would directly, and negatively, affect non-AI children in these counties. As noted in an earlier response, economic prospects of non-AI persons (i.e., employment opportunities, median income) rose over time following the introduction of the casino.

We, instead, interpret the result within the context of two national trends. First, based on a recent report by the National Center for Education Statistics (2022), standardized reading and math achievement scores decreased nationally between 2012 and 2020. The narrowing of the gap shown in Figures 3 and 4 may, in part, reflect a trend in decreasing math and reading scores at the population level over this time period. Second, rural schools (relative to urban and suburban schools) generally show lower reading and math proficiency. Logan and Burdick-Will (2017), for instance, find that among white children of elementary school age in the US, the lowest reading and math proficiency scores occur in rural schools (vs. urban and suburban schools). Drescher and colleagues (2022), moreover, find that, for both white and AI students, 3rd graders in rural districts score about half a grade level lower (relative to their nonrural counterparts). These literatures, however, are limited in that they do not reach agreement regarding causes of (i.) declining national trends and (ii.) these large rural/ nonrural test score gaps. That stated, the literature we cite above offers important context for interpreting the AI**duration* coefficient; we now include additional text to this effect in the Discussion. We also now interpret the narrowing of the test score gaps shown in Figures 3 and 4 as reflecting, in part, a trend in decreasing math and reading scores at the population level over this time period, which is attenuated for AI children by the cash transfer.

In addition, although it does not relate directly to test scores, we note that a separate paper by our team examining the potential benefits to the AI population, following increased *duration* of exposure to the cash transfer as a child, now appears in the literature. We now cite this paper which shows salutary associations between the AI**duration* exposure variable and several measures of maternal (e.g., body mass index) and infant health (healthy birth weight). This evidence supports the plausibility of benefits to the offspring of AI recipients as well as maternal and infant health mechanisms that may link the AI**duration* exposure to better test scores of AI children.

The Reviewer also raises the question of whether measurement issues with the test score data could affect inference. To assess this notion, we conducted several checks. First,

given that the difficulty of the test could vary from year-to-year, we converted all scores to their Z-score (i.e., normed to the mean and standard deviation of the score for that particular calendar year). This Z-score conversion permits comparison across years of relative “rank” in test score achievement. Second, in a separate sensitivity check, we control for test year (both as a continuous variable and with year fixed-effects) in a separate regression. Inference remains very similar to the original tests, which indicates that any temporal pattern in test scores does not drive results (shown again, below, in Supplemental Tables 8 through 11). Third, the falsification check regarding the AI**pre-treatment* interaction term (shown in the previous round of review) does not show any evidence of deviation from parallel trends in the pre-treatment test scores, which rules out the possibility that pre-treatment trends in test scores drive results.

Another possibility, raised by the Reviewer, involves the notion that 3rd grade test scores may not fully capture academic abilities. We agree that test scores alone seem incomplete in measuring the nuanced and rich concept of academic ability. Ideally, we would have additional school information that more comprehensively captures these abilities. Whereas we do not have access to such variables at this time, we may acquire (through a separate follow-up study) information on subject matter grades, school attendance, and other assessments of student performance. We also point out that, while 3rd grade test scores may have these limitations (as do all standardized tests), these scores remain routinely used in education scholarship and are strongly predictive of student success in terms of high school achievement and graduation rates, including in North Carolina.

In light of the points above, we do not view confounding or measurement validity issues of test scores as plausible explanations for the pattern of results we report. However, we acknowledge that we do not have a satisfying *post hoc* explanation regarding the pattern of AI and non-AI specific results. We now note this issue in the Discussion and recommend several specific avenues of future research in this area.

References:

Bustos, B. *et al.* Family cash transfers in childhood and birthing persons and birth outcomes later in life. *SSM - Population Health* **25**, 101623 (2024).

Drescher, J., Podolsky, A., Reardon, S. F. & Torrance, G. The Geography of Rural Educational Opportunity. *RSF: The Russell Sage Foundation Journal of the Social Sciences* **8**, 123–149 (2022).

Logan, J. R. & Burdick-Will, J. School Segregation and Disparities in Urban, Suburban, and Rural Areas. *The ANNALS of the American Academy of Political and Social Science* **674**, 199–216 (2017).

Goldhaber, D., Wolff, M. & Daly, T. Assessing the accuracy of elementary school test scores as predictors of students’ high school outcomes. *National Center for Analysis of Longitudinal Data in Education Research*. Working Paper 235-0520. (2020).

National Center for Education Statistics. Reading and Mathematics Score Trends. *Condition of Education*. U.S. Department of Education, Institute of Education Sciences. <https://nces.ed.gov/programs/coe/indicator/cnj> (2022).

Supplemental Table 8: Control for test year (continuous). Regression results predicting Third Grade **Math** Z-score for 4,289 children in Jackson, Swain, and Graham Counties, 2008-2017, as a function of AI race/ethnicity, duration of mother’s exposure to family cash transfer as a child, and other covariates.

Variables‡	Model 2	
	coef	95% CI
AI race/ethnicity (AI) (reference = non-AI)	-0.423***	[-0.523 -0.323]
Duration of potential exposure of mother to family cash transfer in childhood (duration) ¹	-0.038****	[-0.048 -0.028]
Interaction of AI * duration	0.025**	[0.005 0.045]
Test year (continuous)	-0.008	[-0.018 0.003]

¹ This variable indicates number of years of possible receipt of cash transfer for AI participants; for non-AI participants, it represents an age control.

‡ Child age and gender controls included in regression but not shown here (full results for covariates available in data repository).

(*p < 0.1, ** p < 0.05, ***p < 0.01, ****p < 0.001)

Supplemental Table 9: Control for test year (continuous). Regression results predicting Third Grade **Reading** Z-score for 4,254 children in Jackson, Swain, and Graham Counties, 2008-2017, as a function of AI race/ethnicity, duration of mother’s exposure to family cash transfer as a child, and other covariates.

Variables‡	Model 2	
	coef	95% CI
AI race/ethnicity (AI) (reference = non-AI)	-0.493***	[-0.594 -0.392]
Duration of potential exposure of mother to family cash transfer in childhood (duration) ¹	-0.043****	[-0.053 -0.033]
Interaction of AI * duration	0.028***	[0.008 0.049]
Test year (continuous)	0.0006	[-0.010 0.012]

¹ This variable indicates number of years of possible receipt of cash transfer for AI participants; for non-AI participants, it represents an age control.

‡ Child age and gender controls included in regression but not shown here (full results for covariates available in data repository).

(*p < 0.1, ** p < 0.05, ***p < 0.01, ****p < 0.001)

Supplemental Table 10: Including Test Year Fixed Effects and restricting the analysis to a narrower age range of G2 mothers 16 to 35 years at the time of G3's birth. Regression results predicting Third Grade **Math** Z-score for 3,977 children in Jackson, Swain, and Graham Counties, 2008-2017, as a function of AI race/ethnicity, duration of mother's exposure to family cash transfer as a child, and other covariates.

Variables [‡]	Model 2	
	coef	95% CI
AI race/ethnicity (AI) (reference = non-AI)	-0.376***	[-0.480 -0.272]
Duration of potential exposure of mother to family cash transfer in childhood (duration) ¹	-0.038****	[-0.049 -0.028]
Interaction of AI * duration	0.027**	[0.006 0.048]
Test year fixed effects	Included; not shown	

¹ This variable indicates number of years of possible receipt of cash transfer for AI participants; for non-AI participants, it represents an age control.

[‡] Child age and gender controls included in regression but not shown here (full results for covariates available in data repository).

(*p < 0.1, ** p < 0.05, ***p < 0.01, ****p < 0.001)

Supplemental Table 11: Including Test Year Fixed Effects and restricting the analysis to a narrower age range of G2 mothers 16 to 35 years at the time of G3's birth. Regression results predicting Third Grade **Reading** Z-score for 3,944 children in Jackson, Swain, and Graham Counties, 2008-2017, as a function of AI race/ethnicity, duration of mother's exposure to family cash transfer as a child, and other covariates.

Variables [‡]	Model 2	
	coef	95% CI
AI race/ethnicity (AI) (reference = non-AI)	-0.443***	[-0.549 -0.337]
Duration of potential exposure of mother to family cash transfer in childhood (duration) ¹	-0.040****	[-0.051 -0.030]
Interaction of AI * duration	0.024**	[0.002 0.046]
Test year fixed effects	Included; not shown	

¹ This variable indicates number of years of possible receipt of cash transfer for AI participants; for non-AI participants, it represents an age control.

[‡] Child age and gender controls included in regression but not shown here (full results for covariates available in data repository).

(*p < 0.1, ** p < 0.05, ***p < 0.01, ****p < 0.001)